# REPARAMETERIZATION PROXIMAL POLICY OPTIMIZATION

## ABSTRACT

Reparameterization policy gradient (RPG) is promising for improving sample efficiency by leveraging differentiable dynamics. However, a critical barrier is its training instability, where high-variance gradients can destabilize the learning process. To address this, we draw inspiration from Proximal Policy Optimization (PPO), which uses a surrogate objective to enable stable sample reuse in the model-free setting. We first establish a connection between this surrogate objective and RPG, which has been largely unexplored and is non-trivial. Then, we bridge this gap by demonstrating that the reparameterization gradient of a PPO-like surrogate objective can be computed efficiently using backpropagation through time. Based on this key insight, we propose Reparameterization Proximal Policy Optimization (RPO), a stable and sample-efficient RPG-based method. RPO enables stable sample reuse over multiple epochs by employing a policy gradient clipping mechanism tailored for RPG. It is further stabilized by Kullback-Leibler (KL) divergence regularization and remains fully compatible with existing variance reduction methods. We evaluate RPO on a suite of challenging locomotion and manipulation tasks, where experiments demonstrate that our method achieves superior sample efficiency and strong performance.

## 1 INTRODUCTION

Reparameterization policy gradient (RPG) (Mohamed et al., 2020; Amos et al., 2021) is a model-based method that computes the policy gradient using the reparameterization trick (Kingma & Welling, 2014; Rezende et al., 2014). Different from model-free policy gradients such as REIN-FORCE (Williams, 1992; Sutton et al., 1999), RPG directly backpropagates through the trajectory to obtain a policy gradient estimate. This approach has become increasingly attractive with the recent rise of differentiable simulators (Hu et al., 2020; Freeman et al., 2021; Xu et al., 2021; Xing et al., 2025) and learned world models (Hafner et al., 2020; Amos et al., 2021; Parmas et al., 2023).

RPG has the advantage of exploiting the underlying dynamical structure of the sampling path and can therefore have lower variance and a more accurate policy gradient estimate (Mohamed et al., 2020). However, it can suffer from the exploding/vanishing gradient problem, particularly in environments with non-smooth dynamics or on long-horizon trajectories (Metz et al., 2021; Suh et al., 2022). Previous works have sought to mitigate these issues. A representative method, Short-Horizon Actor-Critic (SHAC) (Xu et al., 2021), reduces variance by only backpropagating through a short horizon of the trajectory. Recently, SAPO (Xing et al., 2025) builds upon SHAC by adding entropy regularization to further stabilize policy training.

Yet, even with current state-of-the-art (SOTA) variance reduction methods, we empirically observe that RPG can still suffer from unstable policy training. Specifically, we trained a locomotion policy with SAPO in the Humanoid environment of the differentiable simulator DFlex (Xu et al., 2021; Georgiev et al., 2024). As shown in Figure 1, SAPO suffers from large policy updates (indicated by spikes in Kullback-Leibler divergence) that lead to sudden performance drops, despite incorporating both short-horizon rollouts and entropy regularization. This result indicates that although SHAC and SAPO reduce gradient variance, they lack an explicit mechanism to control the policy update size, which can lead to instability. This instability hinders RPG-based methods from boosting their sample efficiency to their full potential or from effectively incorporating techniques like sample reuse. This

clearly highlights the need for a more effective algorithm to stabilize the training of RPG, thereby significantly improving its sample efficiency.

In this work, we take inspiration from Proximal Policy Optimization (PPO) (Schulman et al., 2017), a model-free algorithm renowned for stabilizing policy training even with multiple sample reuse. This desirable attribute comes from optimizing a its surrogate objective. However, adapting this principle to RPG-based methods is non-trivial. The surrogate objective is typically optimized via REINFORCE-style gradients and its connection to RPG has been largely unexplored.

To bridge this gap, we establish that RPG is naturally connected to a PPO-like surrogate objective via backpropagation through time (BPTT) (Mozer, 1995), allowing us to compute the reparameterization policy gradients efficiently for both on- and off-policy updates.

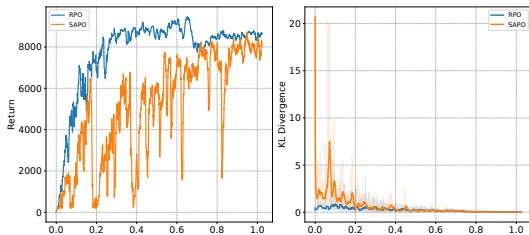

Figure 1: Comparison of RPO and SAPO in the Humanoid environment for the same random seed. **Left:** Training curves showing return versus environment steps. **Right:** The corresponding KL divergence between policy updates, displaying both raw and smoothed curves. SAPO's large KL spikes correspond to sudden performance drops, whereas RPO maintains stable, sample-efficient learning.

This connection is crucial because it provides a principled way to enable stable sample reuse in RPG, thereby stabilizing training and significantly improving its sample efficiency. Based on this insight, we propose **Reparameterization Proximal Policy Optimization (RPO)**. First, it enables sample reuse by optimizing a surrogate objective. To maintain stability, it incorporates a clipped policy gradient mechanism tailored for RPG, which constrains updates arising from large importance weights. Second, it enhances stability with an explicit Kullback-Leibler (KL) divergence regularization term, as we find clipping alone to be insufficient. Furthermore, its practical implementation requires only a single backpropagation pass per trajectory across multiple updates. Finally, RPO is fully compatible with and benefits from existing variance reduction methods for RPG. We conduct experiments on a suite of locomotion and manipulation tasks using two differentiable simulators, DFlex (Xu et al., 2021; Georgiev et al., 2024) and Rewarped (Xing et al., 2025). Experimental results show that RPO achieves superior sample efficiency and strong performance.

In summary, our main contributions are: (i) we show that BPTT can be utilized to compute the on-policy and off-policy reparameterization policy gradients of a PPO-like surrogate objective, which enables stable sample reuse for RPG; (ii) based on this novel insight, we propose RPO, which enables stable and sample-efficient RPG-based policy learning, through multiple sample reuse, an importance weight clipping mechanism, and explicit KL regularization; (iii) we conduct experiments on a suite of locomotion and manipulation tasks using the popular differentiable simulators DFlex and Rewarped. Experiment results clearly demonstrate the superior sample efficiency of RPO and its strong performance.

## 2 RELATED WORK

**Policy Gradient Estimators.** One classical class of policy gradient estimators is based on the score function, such as the REINFORCE gradient estimator (Williams, 1992; Sutton et al., 1999). Many policy gradient methods, such as PPO and TRPO (Schulman et al., 2017; 2015), rely on variants of the REINFORCE gradient estimator. One limitation of the REINFORCE gradient is its high variance, which results in low sample efficiency. On the other hand, if one has access to the underlying dynamic model, either through differentiable simulators (Freeman et al., 2021; Xu et al., 2021; Hu et al., 2020; Xing et al., 2025) or learned world models (Hafner et al., 2020; Amos et al., 2021), another type of policy gradient named Reparameterization Policy Gradient (RPG), which is based on the reparameterization trick (Kingma & Welling, 2014; Rezende et al., 2014), can be obtained. Using the reparameterization trick (Kingma & Welling, 2014; Rezende et al., 2014), RPG directly backpropagates through the trajectory and obtains an unbiased estimate of the policy gradient. By contrast, the REINFORCE gradient estimator does not need to backpropagate through the entire computational graph and only relies on local computation (Parmas, 2018). Since RPG utilizes the

gradients of the dynamics model, RPG typically enjoys less variance than the REINFORCE gradient estimator (Mohamed et al., 2020).

**Reparameterization Policy Gradient-based Reinforcement Learning Algorithms.** It is well known that RPG obtained by vanilla backpropagation through time over a long time horizon suffers from the vanishing/exploding gradient problem (Suh et al., 2022; Metz et al., 2021; Zhang et al., 2023b; Ma et al., 2024). This phenomenon is amplified when dealing with stiff dynamics, such as contact (Zhang et al., 2023a; Suh et al., 2022; Zhong et al., 2023; Pang et al., 2023). RPG can exhibit a large variance when the gradient magnitude is large, which renders the underlying reinforcement learning algorithm unstable, struggling with non-convex loss landscapes.

Several works (Parmas et al., 2018; 2023; Suh et al., 2022) weight and combine RPG and REIN-FORCE according to their variance, while AGPO (Gao et al., 2024) further combines RPG with gradients of Q-functions. SHAC and AHAC (Xu et al., 2021; Georgiev et al., 2024) reduce the variance of RPG by only backpropagating through a truncated length of the trajectory, aided by a value function to estimate future returns. MB-MIX (Zhang et al., 2025) backpropagates a mixture of trajectories with different lengths to better balance the bias-variance trade-off. GI-PPO (Son et al., 2023) first optimizes the policy using RPG, then uses the REINFORCE gradient to perform further off-policy updates in the PPO style. However, the gradients computed by REINFORCE are not only of lower quality than those from RPG, but can also conflict with each other, thereby degrading sample efficiency and performance. Entropy is also introduced to regularize RPG-based policy updates and promote exploration (Xing et al., 2025; Amos et al., 2021).

## 3 PRELIMINARIES

### 3.1 REINFORCEMENT LEARNING FORMULATION

In this work, we consider problems formulated as a Markov Decision Process (MDP) (Sutton & Barto, 2018). An MDP is formally defined by a tuple $(\mathcal{S}, \mathcal{A}, p, r, p_0, \gamma)$, where $\mathcal{S}$ is the set of states, $\mathcal{A}$ is the set of actions, $p : \mathcal{S} \times \mathcal{A} \times \mathcal{S} \to [0, 1]$ is the state transition probability function, $r : \mathcal{S} \times \mathcal{A} \to \mathbb{R}$ is the reward function, $s_0$ is the initial state, $p_0(s_0)$ is the initial state distribution, and $\gamma \in [0, 1)$ is the discount factor.

The goal of reinforcement learning (RL) is to find the optimal parameter $\theta^*$ for a parameterized stochastic policy $\pi_\theta$. A parameterized stochastic policy $\pi_\theta(a|s)$ specifies the probability distribution over actions $a \in \mathcal{A}$ given a state $s \in \mathcal{S}$. The optimal parameter $\theta^*$ maximizes the expected discounted cumulative reward:

$$\theta^* = \arg\max_\theta J(\theta) = \arg\max_\theta \mathbb{E}_{\tau \sim \pi_\theta}[R(\tau)] \tag{1}$$

where the expectation is over trajectories $\tau = (s_0, a_0, s_1, a_1, \dots)$ generated by following the policy $\pi_\theta$. Here, $r(s_t, a_t)$ denotes the reward received at time step $t$, and $R(\tau) = \sum_{t=0}^\infty \gamma^t r(s_t, a_t)$ is the discounted cumulative reward for the trajectory $\tau$.

### 3.2 REPARAMETERIZATION POLICY GRADIENT

RPG relies on the reparameterization trick (Kingma & Welling, 2014; Rezende et al., 2014; Haarnoja et al., 2018) to sample an action $a_t$ from a Gaussian policy $\pi_\theta(a_t|s_t)$. This is achieved by first using the policy network to predict the mean $\mu_\theta(s_t)$ and standard deviation $\sigma_\theta(s_t)$, and then combining them with a reparameterization noise $\epsilon$ at time step $t$:

$$a_t = \mu_\theta(s_t) + \sigma_\theta(s_t) \cdot \epsilon, \quad \text{where } \epsilon \sim \mathcal{N}(0, \mathcal{I}). \tag{2}$$

We denote this entire reparameterization transformation as $a_t = f_\theta(\epsilon; s_t)$.

We focus on deterministic system dynamics $s_{t+1} = g(s_t, a_t)$. Consistent with prior approaches (Xu et al., 2021; Georgiev et al., 2024; Xing et al., 2025), we adopt the following assumption regarding the dynamics and reward function to guarantee well-defined reparameterization policy gradients.

**Assumption 1.** The system dynamics $g(s, a)$ and the reward function $r(s, a)$ are continuously differentiable with respect to state $s$ and action $a$.

With the reparameterization trick, RPG can backpropagate through trajectory by computing Jacobians, $\frac{\partial s_{t+1}}{\partial a_t}$ and $\frac{\partial s_{t+1}}{\partial s_t}$, to obtain a policy gradient estimate:

$$\nabla_\theta J(\theta) = \mathbb{E}_{s_0, \epsilon_0, s_1, \epsilon_1, \ldots} \left[ \nabla_\theta R(\tau) \right]. \tag{3}$$

Note that the expectation is taken with respect to the initial state distribution, sampled noises, and transition dynamics.

However, this full backpropagation through long trajectories often leads to high gradient variance and unstable training. Short-Horizon Actor-Critic (SHAC) (Xu et al., 2021), addresses this by backpropagating through only a short horizon of the trajectory, using a value function to capture the long-term return. SHAC's variant of RPG has the following form:

$$\nabla_\theta [R(\tau_{t_0:t_0+h-1}) + \gamma^h V(s_{t_0+h})], \tag{4}$$

where $t_0$ is the starting time step for the trajectory, $h$ is the short horizon, $R(\tau_{t_0:t_0+h-1})$ is the cumulative reward of the trajectory within the short horizon, and $V(s_{t_0+h})$ is the value function's estimate of the future return.

### 3.3 SURROGATE OBJECTIVE

PPO (Schulman et al., 2017) and TRPO (Schulman et al., 2015) optimize variants of the following surrogate objective function (Kakade, 2001):

$$L_{\pi_{\theta_{\text{old}}}}(\theta) = \int_s \sum_{t=0}^{\infty} \gamma^t p(s_t = s | \pi_{\theta_{\text{old}}}) \int_a A^{\pi_{\theta_{\text{old}}}}(s, a) \pi_\theta(a|s) \mathrm{d}a \mathrm{d}s \tag{5}$$

where $\pi_{\theta_{\text{old}}}$ is the behavior policy used to collect samples; $\sum_{t=0}^{\infty} \gamma^t p(s_t = s | \pi_{\theta_{\text{old}}})$ is the unnormalized state probability density induced by $\pi_{\theta_{\text{old}}}$; and $A^{\pi_{\theta_{\text{old}}}}(s, a)$ is the advantage function corresponding to the behavior policy. This objective measures the performance of the new policy $\pi_\theta$, using the state distribution and advantages for the behavior policy. PPO optimizes a clipped variant of this objective, while TRPO optimizes an explicit KL-constrained variant (Schulman et al., 2017; 2015).

By the law of the unconscious statistician (Grimmett & Stirzaker, 2001), we can rewrite the surrogate objective in the reparameterization form:

$$L_{\pi_{\theta_{\text{old}}}}(\theta) = \int_s d^{\pi_{\theta_{\text{old}}}}(s) \int_\epsilon A^{\pi_{\theta_{\text{old}}}}(s, a)|_{a=f_\theta(\epsilon; s)} p_{\text{standard}}(\epsilon) \mathrm{d}\epsilon \mathrm{d}s \tag{6}$$

where $d^{\pi_{\theta_{\text{old}}}}(s)$ denotes the unnormalized state density induced by $\pi_{\theta_{\text{old}}}$ and $p_{\text{standard}}(\epsilon)$ is the probability density function for standard Gaussian distribution. This surrogate objective differs fundamentally from the objective of the stochastic value gradient (SVG) (Heess et al., 2015; Amos et al., 2021), as Equation 6 measures performance using the value function of the behavior policy, while SVG uses the on-policy value function.

## 4 FROM THEORETICAL INSIGHTS TO ALGORITHMIC DESIGN: REPARAMETERIZATION PROXIMAL POLICY OPTIMIZATION

In this section, we introduce our proposed method: Reparameterization Proximal Policy Optimization (RPO). To achieve stable sample reuse, RPO incorporates three key mechanisms: (i) optimizing the PPO-like surrogate objective via RPG, (ii) a policy gradient clipping mechanism designed for RPG, and (iii) an explicit KL regularization term.

### 4.1 SURROGATE OBJECTIVE FOR POLICY IMPROVEMENT

SOTA RPG methods, e.g., (Xu et al., 2021; Xing et al., 2025), are limited to a single policy update per batch, underutilizing expensive BPTT gradients. While GI-PPO (Son et al., 2023) attempts sample reuse, its hybrid REINFORCE approach could introduce update conflicts and underutilize the low-variance RPG gradient. We establish a novel connection between the PPO-style surrogate objective (6) and the reparameterization gradient. Our key technical contribution leverages this link,

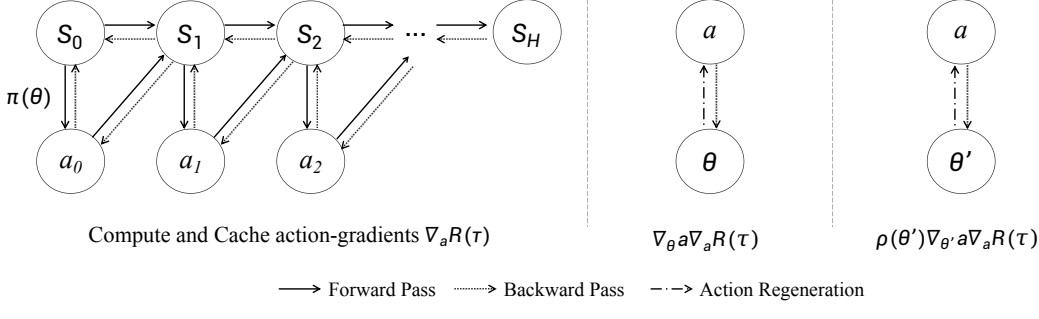

Compute and Cache action-gradients $\nabla_a R(\tau)$      $\nabla_\theta a \nabla_a R(\tau)$      $\rho(\theta') \nabla_{\theta'} a \nabla_a R(\tau)$

$\longrightarrow$ Forward Pass    $\cdots\cdots\triangleright$ Backward Pass    $-\cdot\rightarrow$ Action Regeneration

(a) Rollout and compute action-gradients      (b) On-policy gradient    (c) Off-policy gradient

Figure 2: Computing the reparameterization policy gradient of the surrogate objective involves three steps: **(a)** Action-gradients are computed from rollouts via a single backward pass and cached. **(b)** These gradients are used directly for the initial, on-policy update. **(c)** For subsequent off-policy updates, the cached gradients are importance-weighted by $\rho(\theta')$ and reused, enabling stable sample reuse.

using a single BPTT pass to efficiently compute the RPG gradient for multiple on- and off-policy updates.

The first major component for RPO is the surrogate objective for policy improvement:

$$L_{\pi_{\theta_{\text{old}}}}(\theta) = \mathbb{E}_{s \sim d^{\pi_{\theta_{\text{old}}}}, \epsilon \sim p_{\text{standard}}} \left[ A^{\pi_{\theta_{\text{old}}}}(s, f_\theta(\epsilon; s)) \right] \tag{7}$$

where $\pi_{\theta_{\text{old}}}$ is the policy before the update, $d^{\pi_{\theta_{\text{old}}}}(s)$ denotes the unnormalized state density induced by $\pi_{\theta_{\text{old}}}$, and $\epsilon$ is sampled from the standard Gaussian distribution. Our goal is to compute the reparameterization gradient for the surrogate objective via BPTT. The reparameterization gradient of the surrogate objective with respect to policy parameter $\theta$ has the following form:

$$\nabla_\theta L_{\pi_{\theta_{\text{old}}}}(\theta) = \int_s d^{\pi_{\theta_{\text{old}}}}(s) \int_\epsilon \left[ \nabla_\theta a \nabla_a A^{\pi_{\theta_{\text{old}}}}(s, a)|_{a=f_\theta(\epsilon;s)} p_{\text{standard}}(\epsilon) \right] \mathrm{d}\epsilon \mathrm{d}s$$

$$= \int_s d^{\pi_{\theta_{\text{old}}}}(s) \int_\epsilon \left[ \nabla_\theta a \nabla_a Q^{\pi_{\theta_{\text{old}}}}(s, a)|_{a=f_\theta(\epsilon;s)} p_{\text{standard}}(\epsilon) \right] \mathrm{d}\epsilon \mathrm{d}s. \tag{8}$$

Here $A^{\pi_{\theta_{\text{old}}}}(s, a) = Q^{\pi_{\theta_{\text{old}}}}(s, a) - V^{\pi_{\theta_{\text{old}}}}(s)$ and $V^{\pi_{\theta_{\text{old}}}}(s)$ is constant with respect to the parameter $\theta$ being optimized. Note that $\nabla_a A^{\pi_{\theta_{\text{old}}}}(s, a) = \nabla_a Q^{\pi_{\theta_{\text{old}}}}(s, a)$, since the value function $V^{\pi_{\theta_{\text{old}}}}(s)$ does not depend on actions. Also, equation (8) is rewritten for taking expecation over the whole trajectory as equation (15) in Appendix D.

### 4.1.1 DERIVING THE REPARAMETERIZATION POLICY GRADIENT FOR THE SURROGATE OBJECTIVE

In this section, we show the fundamental connection between RPG and the surrogate objective, which allows us to compute the reparameterization gradient of the surrogate objective via BPTT (More details for this derivation are given in Appendix D).

As the first step, we collect a batch of rollouts with the behavior policy, setting $\pi_{\theta_{\text{old}}} = \pi_\theta$. To illustrate the connection, we consider the resulting infinite-horizon computational graph. We consider the gradients of the discounted cumulative return with respect to the action at each time step. From here, we clearly see that the action-gradient for time step $k$, $\nabla_{a_k} R(\tau) = \gamma^k \nabla_{a_k} \sum_{t=k}^\infty \gamma^{(t-k)} r(s_t, a_t)$, is exactly an unbiased Monte Carlo estimate of $\gamma^k \nabla_a Q^{\pi_{\theta_{\text{old}}}}(s_k, a_k)$, where $s_k$ and $a_k$ are sampled according to $p(s_k = s | \pi_{\theta_{\text{old}}})$ and $\pi_{\theta_{\text{old}}}(a_k | s_k)$. This holds because the reparameterization trick allows us to express $\gamma^k \nabla_a Q^{\pi_{\theta_{\text{old}}}}(s_k, a_k)$ as the expected gradient of the returns with respect to $a_k$ across all possible paths sampled via the reparameterization noise (as we assume deterministic dynamics). As shown in Figure 2 (a), we cache these action-gradients for subsequent calculations.

**On-policy reparameterization policy gradient:** The behavior policy $\pi_{\theta_{\text{old}}}$ is exactly the current policy $\pi_\theta$ being updated during the first policy update epoch. We compute the on-policy gradient by backpropagating the cached action-gradients through the policy network $\theta$, as depicted in Figure 2 (b). Summing over different time steps and taking the average over the batch of trajectories, give an unbiased Monte Carlo estimate of the surrogate objective's on-policy gradient: $\int_s d^{\pi_{\theta_{\text{old}}}}(s) \int_\epsilon \left[ \nabla_\theta a \nabla_a Q^{\pi_{\theta_{\text{old}}}}(s, a)|_{a=f_\theta(\epsilon;s)} p_{\text{standard}}(\epsilon) \right] \mathrm{d}\epsilon \mathrm{d}s$. This holds true because: (i) $s_k$ is sampled according to $p(s_k = s | \pi_{\theta_{\text{old}}})$, (ii) $a_k$ is generated via $a_k = f_{\theta_{\text{old}}}(\epsilon; s_k)$ with $\epsilon \sim p_{\text{standard}}(\epsilon)$, and (iii) $\pi_{\theta_{\text{old}}}$ coincides with $\pi_\theta$ for this first epoch.

**Off-policy reparameterization policy gradient:** The on-policy gradient estimate can be used to update the policy parameters from $\theta$ to $\theta'$. After this update, the behavior policy $\pi_{\theta_{\text{old}}}$ (which generated the data) is now different from the current policy $\pi_{\theta'}$. To perform another update on the same data (i.e., sample reuse), we must compute the off-policy RPG gradient.

We can reuse the exact same cached action-gradients $\nabla_{a_k} R(\tau)$, but we must re-establish a computational path from the new policy parameters $\theta'$ to the action $a_k$ to compute $\nabla_{\theta'} a_k$. We achieve this by regenerating the noise $\epsilon_{\text{reg}}$ that is required for the current policy to produce $a_k$. This allows us to express the action as $a_k = f_{\theta'}(\epsilon_{\text{reg}}; s_k)$, creating a new differentiable path.

As shown in Figure 2 (c), the cached action-gradients are then backpropagated through this new path, yielding an estimate of $\gamma^k \nabla_{\theta'} a_k \nabla_{a_k} Q^{\pi_{\theta_{\text{old}}}}(s_k, a_k)|_{a_k = f_{\theta'}(\epsilon_{\text{reg}}; s_k)}$ for each time step. We further weight each time step's gradient by the importance sampling ratio $\rho(\theta') = \frac{\pi_{\theta'}(a|s)}{\pi_{\theta_{\text{old}}}(a|s)}$, to obtain an unbiased off-policy reparameterization policy gradient estimate for the surrogate objective (a proof for the unbiasedness is given in Proposition 1 in Appendix D).

### 4.1.2 PRACTICAL IMPLEMENTATION FOR POLICY IMPROVEMENT

**Collecting rollouts and computing action-gradients.** Following SHAC (Xu et al., 2021), we collect a batch of $N$ short-horizon trajectories using the current policy $\pi_\theta$. We then utilize BPTT to compute and cache the corresponding action-gradients for each time step, $\nabla_a R(\tau)$. Note that each trajectory in the batch is only backpropagated through once. As we previously discussed, the action-gradient for a specific time step $\nabla_{a_k} R(\tau)$ is an unbiased estimate of $\gamma^k \nabla_a Q^{\pi_{\theta_{\text{old}}}}(s_k, a_k)$. Since $A^{\pi_{\theta_{\text{old}}}}(s_k, a_k) = Q^{\pi_{\theta_{\text{old}}}}(s_k, a_k) - V^{\pi_{\theta_{\text{old}}}}(s_k)$ and $V^{\pi_{\theta_{\text{old}}}}(s_k)$ does not depend on the action $a_k$, $\nabla_{a_k} R(\tau)$ is also an unbiased estimate of $\gamma^k \nabla_a A^{\pi_{\theta_{\text{old}}}}(s_k, a_k)$.

**On-policy and Off-policy Updates.** We perform $M$ optimization epochs on a batch of cached action-gradients. The first update is on-policy, while all subsequent updates ($1 < m \leq M$) are off-policy. Our method for computing the reparameterization policy gradient of the surrogate objective is unified across both cases. For each update step, we perform the following procedure.

First, to compute the gradient for the current policy $\pi_\theta$ using off-policy data, we must re-establish a computational path from policy network parameters $\theta$ to the actions. We achieve this by computing the noise $\epsilon_{\text{reg}}$ that is required for the current policy to regenerate the actions stored in the rollout buffer:

$$\epsilon_{\text{reg}} = f_\theta^{-1}(a; s), \tag{9}$$

where $f_\theta^{-1}(a; s)$ is the inverse of the reparameterization transform. With this recovered noise, we can express the action under the current policy as $a = f_\theta(\epsilon_{\text{reg}}; s)$, which creates a new computational graph connecting the current policy parameters $\theta$ to the action stored in the buffer. Note that SVG Heess et al. (2015) has the same action regeneration mechanism.

Next, we introduce a novel **policy gradient clipping mechanism**, designed specifically for RPG. The proposed policy gradient clipping mechanism serves as a safeguard against numerical instability by filtering out samples with excessive importance weight ratios, thereby preventing action probabilities from becoming critically low.

Unlike PPO's clipping mechanism, our formulation clips the importance weight ratio asymmetrically and does not depend on the sign of the advantage function. This design is crucial because RPG, unlike REINFORCE, does not explicitly increase or decrease the log-likelihood of a specific action.

Specifically, let the importance weight ratio be $\rho(\theta) = \frac{\pi_\theta(a|s)}{\pi_{\theta_{\text{old}}}(a|s)}$. The gradient contribution from this action is non-zero only if $\rho(\theta)$ is within the clipping range, and is weighted by $\rho(\theta)$:

$$\begin{cases} \rho(\theta) \nabla_\theta a \nabla_a R(\tau), & \text{if } 1 - c_{low} \leq \rho(\theta) \leq 1 + c_{high}, \\ 0, & \text{otherwise}, \end{cases} \tag{10}$$

where $\nabla_a R(\tau)$ is the cached action-gradient. By performing this step for all actions in the buffer, we obtain the clipped policy gradient for the surrogate objective.

## 4.2 REGULARIZATION TERMS

While our importance ratio-based policy gradient clipping mechanism is designed for RPG (as detailed above), we find that clipping alone is insufficient to ensure stability. This is partly because the clipping mechanism has no effect during the initial on-policy update, as all importance weight ratios are exactly 1. Hence, we incorporate a KL regularization (Kullback & Leibler, 1951) term, which penalizes large deviations from the behavior policy:

$$L_{KL}(\theta) = \mathbb{E}\left[D_{KL}(\pi_{\theta_{\text{old}}}(\cdot|s) \,||\, \pi_\theta(\cdot|s))\right]. \tag{11}$$

Note that this regularization only takes effect with sample reuse, as the KL divergence and its gradient are zero for the first on-policy update.

We also include an entropy bonus to encourage exploration (Haarnoja et al., 2018; Xing et al., 2025):

$$L_{ent}(\theta) = \mathbb{E}\left[H(\pi_\theta(\cdot|s))\right], \tag{12}$$

where $H(\pi_\theta(\cdot|s))$ denotes the entropy of $\pi_\theta$ at a given state.

## 4.3 OVERALL POLICY TRAINING OBJECTIVE AND POLICY UPDATE

The overall policy training objective is to maximize a weighted combination of all three components:

$$L_{policy}(\theta) = \lambda_{surr}L_{\pi_{\theta_{\text{old}}}}(\theta) - \lambda_{KL}L_{KL}(\theta) + \lambda_{ent}L_{ent}(\theta), \tag{13}$$

where $\lambda_{surrogate}, \lambda_{kl}$ and $\lambda_{ent}$ are the coefficients for the three terms. The three gradient components (from the surrogate objective and the two regularization terms) are combined according to their coefficients, and the final resulting gradient is used to update the policy parameters.

## 4.4 VALUE FUNCTION TRAINING

The value function network is trained by minimizing the following regression loss (Xu et al., 2021):

$$L_\phi = \mathbb{E}\left[||V_\phi(s) - \hat{V}(s)||^2\right], \tag{14}$$

where $V_\phi(s)$ is the estimate of the value function, and $\hat{V}(s)$ is the value target computed by TD-$\lambda$ (Sutton & Barto, 2018). We follow SAPO (Xing et al., 2025) using the double-critic network and including the mean of the two value functions for computing the value target.

---

**Algorithm 1: Reparameterization Proximal Policy Optimization (RPO)**

---

1: Initialize policy parameters $\theta$ and value function parameters $\phi$.
2: **for** iteration $k = 1, 2, \ldots, K$ **do**
3:     Initialize empty buffer $\mathcal{B}$.
4:     Collect a batch of short-horizon trajectories by running policy $\pi_\theta$ in parallel environments and store them in buffer $\mathcal{B}$.
5:     **// Compute and cache action-gradients**
6:     Compute and cache the gradients of the discounted cumulative reward w.r.t. each action: $\nabla_a R(\tau)$.
7:     **for** policy update epochs $m = 1, 2, \ldots, M$ **do**
8:         Regenerate the actions stored in $\mathcal{B}$ (Equation 9) with $\pi_\theta$.
9:         Backpropagate the clipped cached action-gradients to policy network parameters $\theta$, weighted by the importance weight ratios (Equation 10).
10:        Compute the gradients of the KL divergence and entropy regularization terms.
11:        Combine the gradients and update the policy network parameters $\theta$.
12:     **end for**
13:     **for** value update epochs $l = 1, 2, \ldots, L$ **do**
14:        Update value function parameters $\phi$ by minimizing the regression loss (Equation 14).
15:     **end for**
16: **end for**

---

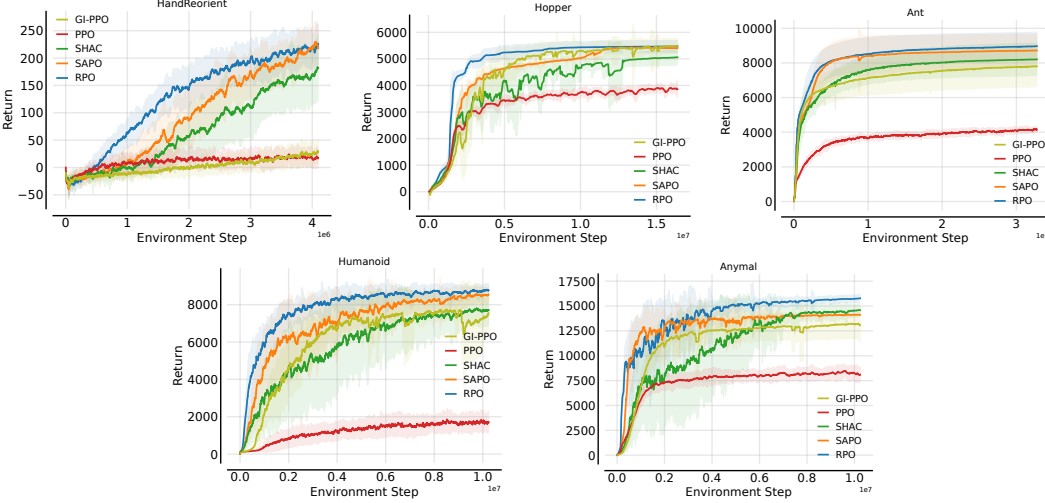

Figure 3: Training performance comparison of RPO, SAPO, SHAC, and PPO. Each plot shows the mean episode return as a function of environment steps, with the shaded region representing the standard deviation. All curves are smoothed with a 100-episode moving average.

## 5 EXPERIMENTS

We conduct experiments to answer the following three questions: (i) Does RPO achieve superior sample efficiency compared to that of previous RPG-based methods? (ii) Does RPO achieve strong performances? (iii) What are the impacts of RPO's different main components on its overall performance?

### 5.1 EXPERIMENTAL SETUP

**Environments and tasks.** We conduct experiments on a suite of five challenging continuous control tasks from two differentiable simulators, DFlex (Xu et al., 2021; Georgiev et al., 2024) and Rewarped (Xing et al., 2025). This suite is composed of four locomotion tasks and one dexterous manipulation task. The four locomotion tasks are from DFlex (Georgiev et al., 2024), where the goal is to maximize the forward velocity: (i) Hopper; (ii) Ant; (iii) Anymal; and (iv) Humanoid. The manipulation task is the Hand Reorient environment from Rewarped (Xing et al., 2025), which involves an Allegro Hand learning to reorient a cube. Further details regarding the environments are provided in Appendix A.

**Baselines.** We compare the sample efficiency and performance of RPO with leading RPG-based and model-free methods: (a) SAPO (Xing et al., 2025), an RPG-based method with short-horizon trajectories and entropy regularization; (b) SHAC (Xu et al., 2021), a variance reduction method for RPG, for which we use the implementation from (Xing et al., 2025) that includes several architectural changes that enhance its performance; (c) PPO (Schulman et al., 2017), a model-free policy gradient method; (d) GI-PPO (Son et al., 2023), which performs a single RPG update epoch followed by PPO-style updates for all subsequent epochs via REINFORCE. We made several architecture improvements to GI-PPO for a fair comparison. Detailed hyper-parameters and implementation specifics for all methods are provided in Appendix C.

**Metrics.** We evaluate each algorithm using 12 random seeds. To account for simulator stochasticity, we run each seed twice (for 24 effective runs per experiment), except in the deterministic Hopper environment. Sample efficiency is evaluated via training curves in Figure 3. We report the final performance over 128 episodes using both deterministic and stochastic protocols in Table 1 and Appendix B, respectively.

### 5.2 EXPERIMENTAL RESULTS

**RPO consistently achieves improved sample efficiency and stability over baselines.** RPO's superior sample efficiency stems from its ability to (i) stabilize RPG policy training via a clipping mechanism and KL regularization, and (ii) reuse samples across multiple policy updates. As shown in Figure 3, this translates to faster learning across tasks. For example, in Hand Reorient (top left),

|  | **Hand Reorient** | **Hopper** | **Ant** | **Humanoid** | **Anymal** |
|---|---|---|---|---|---|
| PPO | $20.88 \pm 15.90$ | $3977.85 \pm 159.95$ | $4339.51 \pm 745.48$ | $2140.61 \pm 529.07$ | $10257.12 \pm 2247.05$ |
| GI-PPO | $36.78 \pm 20.90$ | $\mathbf{5514.00 \pm 285.64}$ | $7812.05 \pm 1165.10$ | $7576.97 \pm 621.58$ | $12247.36 \pm 3552.17$ |
| SHAC | $175.13 \pm 55.10$ | $5068.42 \pm 299.73$ | $8205.95 \pm 940.18$ | $7722.20 \pm 742.96$ | $14560.59 \pm 655.30$ |
| SAPO | $\mathbf{225.17 \pm 27.66}$ | $\mathbf{5478.12 \pm 4.45}$ | $9101.36 \pm 996.14$ | $8676.25 \pm 420.62$ | $14783.27 \pm 53.01$ |
| **RPO (ours)** | $\mathbf{230.15 \pm 33.12}$ | $\mathbf{5534.32 \pm 223.68}$ | $\mathbf{9249.50 \pm 831.56}$ | $\mathbf{8958.41 \pm 373.64}$ | $\mathbf{16006.42 \pm 342.49}$ |

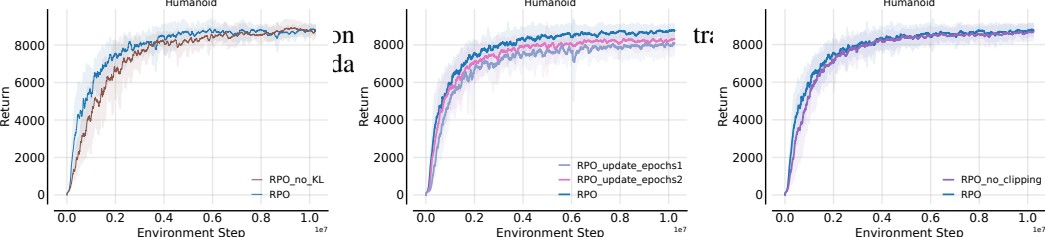

Figure 4: Ablation study of RPO's components in the Humanoid environment. The plot shows training curves for four variants: RPO without KL regularization, RPO with only one or two policy update epochs, and RPO without the importance weight clipping mechanism.

SAPO requires an additional 3 million environment steps to achieve a performance comparable to RPO. In Hopper (top middle), RPO reaches a score of 5000 approximately 5 million steps sooner than all other methods. Similarly, in Ant (top right), RPO is the fastest to reach a score of 6000. RPO surpasses the final performance of both SAPO and SHAC in Anymal (bottom right) after only 4 million steps, and it reaches a score of 8000 in Humanoid (bottom left) around 3 million steps faster than SAPO. Furthermore, RPO is also significantly more sample-efficient than PPO in all tasks. Notably, it demonstrates better training stability than SAPO, SHAC and GI-PPO, particularly in Humanoid, Hopper and Hand Reorient environments.

**RPO achieves strong final performance.** As summarized in Table 1, RPO consistently achieves SOTA performance across all tasks. In the Anymal environment, RPO outperforms all other baselines by a significant margin. In the Humanoid and Ant environments, RPO outperforms SAPO while being significantly better than the other methods. For the Hopper and Hand Reorient tasks, RPO achieves the best performance, on par with SAPO and GI-PPO. In summary, RPO not only improves sample efficiency but also demonstrates strong final performance. Stochastic evaluation results, which follows the same trend, are provided in the Appendix B.

### 5.3 Ablation Study

In our ablation study, we investigate the effect of RPO's three main components. First, to test the necessity of KL regularization, we evaluate a variant of RPO without the KL loss. Second, to isolate the effect of gradient clipping, we evaluate RPO with the clipping mechanism removed. Finally, to measure the benefit of sample reuse, we compare RPO against two variants with only one or two policy update epochs. The results are shown in Figure 4. We also test RPO's sensitivity to hyperparameters, and the results are presented in Appendix J.

(i) **KL regularization stabilizes policy training.** KL regularization is critical for RPO's stable policy training, especially during the early phase. As shown in Figure 4, without KL regularization, policy training experiences large policy updates indicated by large KL spikes. One typical example is shown in Figure 5 (a) and (b). As shown in Table 2, the performance at 1 million environment steps is significantly degraded, which indicates KL regularization is key for high sample efficiency. More ablation studies for KL regularization are shown in Appendix H.

(ii) **Clipping mechanism helps to stabilize policy training.** The purpose of the proposed policy gradient clipping mechanism is to filter out samples with large importance weight ratios to avoid numerical instability and to prevent the probability for a certain action from becoming too low. As shown in Figure 5 (d), we measure the average max importance weight ratios for policy update epochs 2 to 4. The importance weight ratios can become large enough to cause instabilities during the early stage of training, and our proposed clipping mechanism works as a natural safeguard against numerical instability. As shown by the results in Table 2, the clipping mechanism stabilizes training during the early training phase (in the first 2 million environment steps).

| No KL Loss | No Clipping | RPO (Ours) |
|---|---|---|
| 5670.6 (1937.7 – 8175.4) | 6526.9 (888.5 – 8743.7) | **7032.3** (3894.6 – 8134.9) |

Table 2: Ablation study of deterministic performance at 1 million environment steps in the Humanoid environment. We report the mean score with the minimum and maximum range in parentheses.

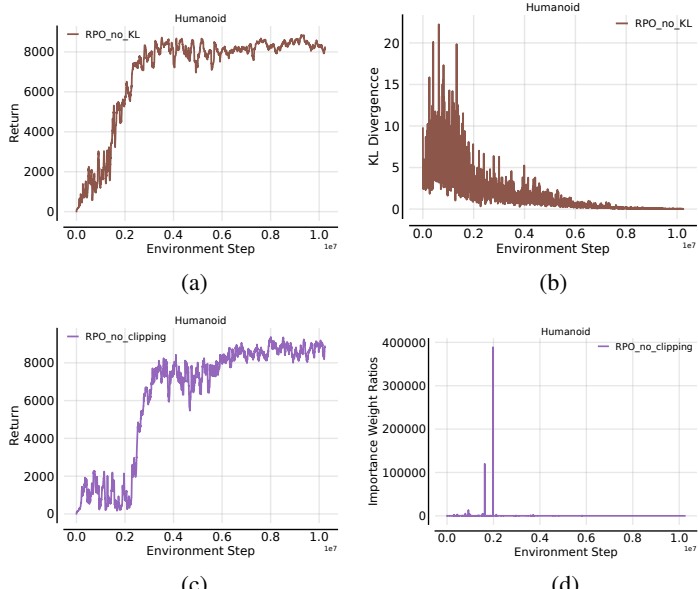

Figure 5: Detailed analysis of RPO without KL regularization and gradient clipping mechanism using representative single seed (to visualize instability that is often smoothed out in aggregate plots). (a) RPO without KL regularization experiences significant instability during the early phase. (b) This instability corresponds to large, uncontrolled spikes in KL divergence. (c) RPO without gradient clipping also suffers from early-phase instability. (d) We measure the average max importance weight ratios for policy update epochs 2 to 4; removing clipping fails to filter out these extreme ratios, directly causing the observed instability.

(iii) **Sample reuse improves sample efficiency.** Sample efficiency degrades when using only one or two policy update epochs (note that the default RPO uses five), and the final performance is also reduced. It is interesting to note that both performance and sample efficiency improve as the number of policy update epochs increases from 1 to 2, and up to 5. Additional ablation analysis for sample reuse on the Hopper task is shown in Appendix I, and the results confirm this trend. These findings highlight the critical role of sample reuse.

## 6 CONCLUSION

In this work, we addressed the training instability of Reparameterization Policy Gradient by establishing a key connection between RPG and a surrogate objective. This insight provides a principled path to stable sample reuse. Based on this, we propose Reparameterization Proximal Policy Optimization (RPO), an algorithm that stabilizes policy learning by applying a tailored gradient clipping mechanism to the surrogate objective's policy gradient, further complemented by KL regularization. Our experiments on challenging locomotion and manipulation tasks confirm that RPO significantly outperforms prior methods in sample efficiency with strong performance. A promising direction for future work is investigating the sim-to-real transfer of RPO-trained policies.

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

# Appendices

## A    ENVIRONMENT AND TASK DETAILS

In this section, we discuss the details of the environments and tasks used in this work. The four locomotion tasks (i.e., Anmal, Hopper, Ant, and Humanoid) are from the DFlex simulator (Xu et al., 2021; Georgiev et al., 2024). Specifically, we use the versions from AHAC's official implementation (https://github.com/imgeorgiev/DiffRL). All locomotion tasks aim to learn a policy that maximizes the agent's forward velocity. The Hand Reorient task is from the official implementation of Rewarped (https://github.com/rewarped/rewarped), version 1.3.0.

### A.1    ANT

Ant ($S \in \mathbb{R}^{37}, A \in \mathbb{R}^8$) is a four-legged robot. The reward function is defined as (Georgiev et al., 2024):
$$v_x + R_{height} + 0.1R_{angle} + R_{heading} - 0.01\|a\|^2,$$
where $v_x$ is the forward velocity, and the other components are: $R_{height}$, which encourages the robot to stand up; $R_{angle}$, which rewards an upward-pointing normal vector; $R_{heading}$, which promotes forward movement; and a penalty on the action norm, $-0.01\|a\|^2$, to encourage energy-efficient policies.

### A.2    ANYMAL

Anymal ($S \in \mathbb{R}^{49}, A \in \mathbb{R}^{12}$) is a real quadrupedal robot (Hutter et al., 2017). The reward function is defined as (Georgiev et al., 2024):
$$v_x + R_{height} + 0.1R_{angle} + R_{heading} - 0.01\|a\|^2.$$

### A.3    HOPPER

Hopper ($S \in \mathbb{R}^{11}, A \in \mathbb{R}^3$) is a three-jointed planar robot. The reward function is defined as (Georgiev et al., 2024):
$$v_x + R_{height} + R_{angle} - 0.1\|a\|^2.$$

### A.4    HUMANOID

Humanoid ($S \in \mathbb{R}^{76}, A \in \mathbb{R}^{21}$) is a high-dimensional bipedal robot. The reward function is defined as (Georgiev et al., 2024):
$$v_x + R_{height} + 0.1R_{angle} + R_{heading} - 0.02\|a\|^2.$$

### A.5    HAND REORIENT

This task involves an Allegro Hand ($S \in \mathbb{R}^{72}, A \in \mathbb{R}^{16}$) learning to reorient a cube to a target pose. This task was adapted for Rewarped (Xing et al., 2025) from Isaac Gym (Makoviychuk et al., 2021). The detailed reward function can be found in the original Isaac Gym paper (Makoviychuk et al., 2021).

## B    STOCHASTIC EVALUATION RESULTS

In this section, we provide the stochastic evaluation results for the five tasks used in the paper. For stochastic evaluations, we sample actions from the policy distribution. Results are shown in Table 7. As shown in Table 7, RPO achieves best results across tasks.

## C    HYPERPARAMETERS AND IMPLEMENTATION DETAILS

### C.1    HYPERPARAMETERS AND ARCHITECTURES

We detail the hyperparameters and architectures used for all algorithms in Table 3. For our GI-PPO baseline, we followed the official implementation (https://github.com/SonSang/gippo) but made

| | *shared* | PPO | SHAC | SAPO | RPO | GI-PPO |
|---|---|---|---|---|---|---|
| Horizon $H$ | 32 | | | | | |
| Epochs for critics $L$ | | 5 | 16 | 16 | 32 | 16 |
| Epochs for actors $M$ | | 5 | 1 | 1 | 5 | 6 |
| Discount $\gamma$ | 0.99 | | | | | |
| TD/GAE $\lambda$ | 0.95 | | | | | |
| Actor MLP | $(400, 200, 100)$ | shared actor-critic MLP | | | | |
| Critic MLP | $(400, 200, 100)$ | shared actor-critic MLP | | | | |
| Actor $\eta$ | | $5e-4$ | $2e-3$ | $2e-3$ | $5e-4$ | $5e-4$ |
| Critic $\eta$ | $5e-4$ | | | | | |
| Entropy $\eta$ | - | | | $5e-3$ | | |
| $\eta$ schedule | - | KL(0.008) | linear | linear | exponential | N.A. |
| Optim type | AdamW | | | | | |
| Optim $(\beta_1, \beta_2)$ | $(0.7, 0.95)$ | $(0.9, 0.999)$ | | | | |
| Grad clip | 0.5 | | | | | 1.0 |
| Norm type | LayerNorm | | | | | |
| Activation type | SiLU | | | | | |
| Num critics $C$ | - | | 2 | 2 | 2 | 2 |
| Target entropy $\bar{\mathcal{H}}$ | - | | | $-\dim(\mathcal{A})/2$ | $-\dim(\mathcal{A})/2$ | |
| Init temperature | - | | | 1.0 (0.005 for Hand Reorient) | | |

Table 3: Common hyperparameters for all algorithms.

| | Hopper | Ant | Humanoid | Anymal | Hand Reorient |
|---|---|---|---|---|---|
| Num Envs | 1024 | 128 | 64 | 128 | 64 |

Table 4: The number of parallel environments used for each environment. These values are kept the same as in the official implementations: we follow the AHAC repository (https://github.com/imgeorgiev/DiffRL) for the DFlex tasks and the Rewarped repository (https://github.com/rewarped/rewarped) for the Hand Reorient task.

| | Hopper | Ant | Humanoid | Anymal | Hand Reorient |
|---|---|---|---|---|---|
| Entropy coefficient | 0.25 | 0.35 | 0.5 | 0.25 | 0.001 |
| KL coefficient | 0.2 | 0.5 | 0.5 | 0.2 | 0.003 |
| $c_{low}$ | 0.8 | 0.8 | 0.8 | 0.8 | 0.8 |
| $c_{high}$ | 1.0 | 1.0 | 1.0 | 1.0 | 1.0 |

Table 5: RPO's unique hyperparameters.

| | Hopper | Ant | Humanoid | Anymal | Hand Reorient |
|---|---|---|---|---|---|
| alpha | $5e-1$ | $5e-1$ | $5e-3$ | $1e-3$ | $5e-4$ |
| max oorr | 0.7 | 0.8 | 0.1 | 0.5 | 0.1 |
| e clip | 0.2 | 0.2 | 0.2 | 0.2 | 0.05 |
| alpha interval | 0.4 | 0.4 | 0.4 | 0.4 | 0.4 |
| alpha update factor | 1.02 | 1.02 | 1.02 | 1.02 | 1.02 |

Table 6: GI-PPO's unique hyperparameters.

| | **Hand Reorient** | **Hopper** | **Ant** | **Humanoid** | **Anymal** |
|---|---|---|---|---|---|
| PPO | $17.03 \pm 14.83$ | $3940.60 \pm 129.72$ | $4146.00 \pm 164.14$ | $1665.66 \pm 410.93$ | $8244.86 \pm 680.87$ |
| GI-PPO | $27.19 \pm 20.58$ | $\mathbf{5512.30 \pm 285.90}$ | $7805.47 \pm 1159.23$ | $7507.85 \pm 639.52$ | $12260.45 \pm 3650.35$ |
| SHAC | $174.63 \pm 57.54$ | $5067.18 \pm 299.37$ | $8206.15 \pm 940.46$ | $7744.44 \pm 858.97$ | $14568.97 \pm 652.72$ |
| SAPO | $\mathbf{213.44 \pm 33.95}$ | $5407.79 \pm 4.28$ | $8718.60 \pm 946.94$ | $8603.09 \pm 402.82$ | $14095.90 \pm 82.02$ |
| RPO (ours) | $\mathbf{214.36 \pm 32.12}$ | $\mathbf{5469.33 \pm 226.29}$ | $\mathbf{8966.00 \pm 817.82}$ | $\mathbf{8767.16 \pm 371.01}$ | $\mathbf{15776.12 \pm 343.17}$ |

Table 7: Stochastic Evaluation (i.e. sampling actions from the policy distribution) for the final performance after training. Each evaluation consists of 128 episodes. Mean and standard deviation.

several improvements. These changes include using a double critic architecture, switching from the Adam to the AdamW optimizer, aligning the network size with our method, and changing the activation function from ELU to SiLU. Since GI-PPO is sensitive to its hyperparameters, we performed an extensive search within our computational budget. The resulting hyperparameters used are listed in Table 6. Our implementations of SAPO, PPO, and SHAC are based on the official SAPO repository (https://github.com/etaoxing/mineral). Most hyperparameters are kept consistent with that repository, with a few key exceptions for fair comparison: the number of parallel environments and the MLP size are aligned with the official AHAC repository (Georgiev et al., 2024). To ensure a fair comparison, most hyperparameters and the core architecture are shared across all tested algorithms. We tuned the initial temperature for SAPO in the Hand Reorient task, as the original setting of 1.0 from the SAPO paper was found to be too high. Specifically for SHAC, we use the improved version from the SAPO repository, which aligns its architecture with that of RPO and SAPO. Our RPO implementation is also built upon the SAPO repository.

### C.2 IMPLEMENTATION DETAILS FOR RPO

For the critic, we use double critic and mean average as target for TD training, following (Xing et al., 2025). For the entropy regularization, we follow SAPO to add an entropy bonus to the reward, which is scaled by a target entropy (Xing et al., 2025). But note that the gradients of entropy are not backpropagated to actions at the same time step, so we calculate the gradients of entropy explicitly with respect to the policy parameter during policy update. We do not discount the explicitly calculated entropy gradients (but the gradients of entropy backpropagated to previous time steps are discounted), and this works empirically well. For the KL divergence computation, as we transform gaussian distribution to squashed normal distribution with tanh, and KL is invariant under such transformation. Hence, we could utilize the closed form expression for KL between gaussian to calculate the gradient of KL.

## D DETAILS AND PROOFS FOR CONNECTING RPG AND SURROGATE OBJECTIVE

In this section, we give details to explain the connection between RPG and surrogate objective. First, we rewrite Equation 8 by expanding $d^{\pi_{\theta_{\text{old}}}}(s)$ and interchanging the order of integration and summation:

$$
\begin{aligned}
\nabla_\theta L_{\pi_{\theta_{\text{old}}}}(\theta) &= \int_s d^{\pi_{\theta_{\text{old}}}}(s) \int_\epsilon \left[ \nabla_\theta a \nabla_a Q^{\pi_{\theta_{\text{old}}}}(s,a)|_{a=f_\theta(\epsilon;s)} p_{\text{standard}}(\epsilon) \mathrm{d}\epsilon \right] \mathrm{d}s, \\
&= \int_s \sum_{t=0}^\infty \gamma^t p(s_t = s | \pi_{\theta_{\text{old}}}) \int_\epsilon \left[ \nabla_\theta a \nabla_a Q^{\pi_{\theta_{\text{old}}}}(s,a)|_{a=f_\theta(\epsilon;s)} p_{\text{standard}}(\epsilon) \mathrm{d}\epsilon \right] \mathrm{d}s, \\
&= \sum_{t=0}^\infty \int_s \gamma^t p(s_t = s | \pi_{\theta_{\text{old}}}) \int_\epsilon \left[ \nabla_\theta a \nabla_a Q^{\pi_{\theta_{\text{old}}}}(s,a)|_{a=f_\theta(\epsilon;s)} p_{\text{standard}}(\epsilon) \mathrm{d}\epsilon \right] \mathrm{d}s,
\end{aligned}
\tag{15}
$$

which is the policy gradient to be estimated and $p_{\text{standard}}(\epsilon)$ is the probability density function for standard Gaussian distribution.

### D.1 COLLECT ROLLOUTS AND COMPUTE ACTION-GRADIENTS

First, as shown in Section 4.1.1, we collect a batch of rollouts and compute the gradients of discounted cumulative return with respect to the action at each time step with BPTT. From here, we clearly see that the action-gradient for time step $k$, $\gamma^k \nabla_{a_k} \sum_{t=k}^\infty \gamma^{(t-k)} r(s_t, a_t)$, is exactly an unbiased Monte Carlo estimate of $\gamma^k \nabla_a Q^{\pi_{\theta_{\text{old}}}}(s_k, a_k)$, where $s_k$ and $a_k$ are sampled according to $p(s_k = s | \pi_{\theta_{\text{old}}})$ and $\pi_{\theta_{\text{old}}}(a|s)$. We cache these action-gradients for further calculations.

### D.2 ON-POLICY GRADIENT

Now, we are ready to compute the reparameterization gradients. For the first policy update (on-policy update), the behavior policy $\pi_{\theta_{\text{old}}}$ is the same as the policy $\pi_\theta$ being updated. We can backpropagate

the gradients from the action to the policy network parameters $\theta$. Then, we get an unbiased Monte Carlo estimate of $\int_s \gamma^k p(s_k = s|\pi_{\theta_{\text{old}}}) \int_\epsilon \left[ \nabla_\theta a \nabla_a Q^{\pi_{\theta_{\text{old}}}}(s,a)|_{a=f_\theta(\epsilon;s)} p_{\text{standard}}(\epsilon) \right] \text{d}\epsilon \text{d}s$. This holds true, as $s_k$ is sampled according to $p(s_k = s|\pi_{\theta_{\text{old}}})$, $a_k$ is generated by sampling $\epsilon \sim p_{\text{standard}}(\epsilon)$ and then applying the transformation $a_k = f_{\theta_{\text{old}}}(\epsilon; s_k)$, and $\pi_{\theta_{\text{old}}}$ coincides with $\pi_\theta$ for this policy update epoch.

By summing the gradients over different time steps and averaging across different trajectories, we obtain an unbiased on-policy reparameterization gradient estimation for Equation 15.

### D.3 OFF-POLICY GRADIENT

After the first policy update, $\pi_\theta$ is updated to $\pi_{\theta'}$, and we need to compute off-policy gradients. To do so, we need to account for the fact that to regenerate the same action collected in the rollout, a different sampled noise $\epsilon_{\text{reg}}$ is required for $\theta'$.

To generate the sampled action $a_k$, $\epsilon_{\text{reg}}$ and $\epsilon$ are linked by the following relation:

$$a_k = f_{\theta_{\text{old}}}(\epsilon; s_k) = f_{\theta'}(\epsilon_{\text{reg}}; s_k). \tag{16}$$

Since we consider reparameterization Gaussian transformations in this work, $f_{\theta_{\text{old}}}$ and $f_{\theta'}$ are invertible. We can therefore compute $\epsilon_{\text{reg}} = f_{\theta'}^{-1}(a_k; s_k)$ and then regenerate $a_k$ with the updated policy $\pi_{\theta'}$. Now, we backpropagate cached action-gradients to policy network parameters $\theta'$ and obtain an unbiased estimation of $\gamma^k \nabla_{\theta'} a_k \nabla_{a_k} Q^{\pi_{\theta_{\text{old}}}}(s_k, a_k)|_{a_k = f_{\theta'}(\epsilon_{\text{reg}}; s_k)}$.

However, instead of sampling $\epsilon_{\text{reg}}$ directly from $\mathcal{N}(0, \mathcal{I})$, we sample it as $\epsilon_{\text{reg}} = f_{\theta'}^{-1}(f_{\theta_{\text{old}}}(\epsilon; s_k); s_k)$, where $\epsilon$ is sampled from $\mathcal{N}(0, \mathcal{I})$. Hence, the probability density $p_{\text{reg}}(\epsilon_{\text{reg}})$ is different from standard Gaussian. We therefore multiply the computed gradient term $\gamma^k \nabla_{\theta'} a_k \nabla_{a_k} Q^{\pi_{\theta_{\text{old}}}}(s_k, a_k)|_{a_k = f_{\theta'}(\epsilon_{\text{reg}}; s_k)}$ by the importance ratio $\rho(\theta') = \frac{\pi_{\theta'}(a_k|s_k)}{\pi_{\theta_{\text{old}}}(a_k|s_k)}$ to obtain an unbiased off-policy gradient estimate. We now show that weighting by this importance weight ratio, we indeed obtain an unbiased estimate of the target gradient: $\int_s \gamma^k p(s_k = s|\pi_{\theta_{\text{old}}}) \int_\epsilon \left[ \nabla_{\theta'} a \nabla_a Q^{\pi_{\theta_{\text{old}}}}(s,a)|_{a=f_{\theta'}(\epsilon;s)} p_{\text{standard}}(\epsilon) \right] \text{d}\epsilon \text{d}s$.

**Proposition 1.** *Let the state $s_k$ be sampled from the state distribution of the behavior policy, $s_k \sim p(s_k = s|\pi_{\theta_{\text{old}}})$. Let the reparameterization noise $\epsilon$ be sampled from the standard Normal distribution, $\epsilon \sim p_{standard}(\epsilon)$, and define the regenerated noise as $\epsilon_{reg} = f_{\theta'}^{-1}(f_{\theta_{old}}(\epsilon; s_k); s_k)$, where $f_{\theta'}$ and $f_{\theta_{old}}$ are reparameterization Gaussian transformations.*

*Define the off-policy gradient estimator $G(\theta')$ as the random variable:*

$$G(\theta') = \gamma^k \frac{\pi_{\theta'}(a_k|s_k)}{\pi_{\theta_{old}}(a_k|s_k)} \nabla_{\theta'} a_k \nabla_{a_k} Q^{\pi_{\theta_{old}}}(s_k, a_k)|_{a_k = f_{\theta'}(\epsilon_{reg}; s_k)}$$

*where the action $a_k = f_{\theta_{old}}(\epsilon; s_k) = f_{\theta'}(\epsilon_{reg}; s_k)$.*

*Then, $G(\theta')$ is an unbiased estimate of the true policy gradient. That is, its expectation over the distributions of $s_k$ and $\epsilon$ is given by:*

$$\mathbb{E}_{s_k, \epsilon} [G(\theta')] = \int_s \gamma^k p(s_k = s|\pi_{\theta_{old}}) \int_\epsilon \left[ \nabla_{\theta'} a \nabla_a Q^{\pi_{\theta_{old}}}(s_k, a)|_{a=f_{\theta'}(\epsilon;s_k)} p_{standard}(\epsilon) \right] \text{d}\epsilon \text{d}s$$

*Proof.* We first consider the case where the action $a \in \mathbb{R}$ is a scalar, and then generalize the result to the multi-dimensional case.

Since $\epsilon_{\text{reg}}$ is sampled as $\epsilon_{\text{reg}} = f_{\theta'}^{-1}(f_{\theta_{\text{old}}}(\epsilon; s_k); s_k)$, $f_{\theta'}$ and $f_{\theta_{\text{old}}}$ are differentiable and monotonically increasing with respect to $\epsilon_{\text{reg}}$ and $\epsilon$, we can calculate the probability density function $p_{\text{reg}}(\epsilon_{\text{reg}})$ by the change of variable formula (Grimmett & Stirzaker, 2001):

$$p_{\text{reg}}(\epsilon_{\text{reg}}) = p_{\text{standard}}(\epsilon) \frac{\text{d}f_{\theta_{\text{old}}}^{-1}(a_k; s_k)}{\text{d}a_k} \frac{\text{d}f_{\theta'}(\epsilon_{\text{reg}}; s_k)}{\text{d}\epsilon_{\text{reg}}}.$$

We also know that the importance weight ratio has the following specific form:

$$\frac{\pi_{\theta'}(a_k|s_k)}{\pi_{\theta_{\text{old}}}(a_k|s_k)} = \frac{p_{\text{standard}}(\epsilon_{\text{reg}})}{p_{\text{standard}}(\epsilon)} \frac{\frac{\text{d}f_{\theta'}^{-1}(a_k; s_k)}{\text{d}a_k}}{\frac{\text{d}f_{\theta_{\text{old}}}^{-1}(a_k; s_k)}{\text{d}a_k}},$$

since $a_k = f_{\theta_{\text{old}}}(\epsilon; s_k) = f_{\theta'}(\epsilon_{\text{reg}}; s_k)$, where $\epsilon$ and $\epsilon_{\text{reg}}$ are both sampled from standard Gaussian distributions, and applying the change of variable formula.

Now, we take the expectation of $G(\theta')$ with respect to $\epsilon_{\text{reg}}$ with probability density function $p_{\text{reg}}$:

$$\gamma^k \int_{\epsilon_{\text{reg}}} \left[ \frac{\pi_{\theta'}(a_k|s_k)}{\pi_{\theta_{\text{old}}}(a_k|s_k)} \nabla_{\theta'} a \nabla_a Q^{\pi_{\theta_{\text{old}}}}(s, a)|_{a=f_{\theta'}(\epsilon_{\text{reg}};s)} p_{\text{reg}}(\epsilon_{\text{reg}}) \mathrm{d}\epsilon_{\text{reg}} \right]$$

$$= \gamma^k \int_{\epsilon_{\text{reg}}} \left[ \frac{p_{\text{standard}}(\epsilon_{\text{reg}})}{p_{\text{standard}}(\epsilon)} \frac{\frac{\mathrm{d}f_{\theta'}^{-1}(a_k;s_k)}{\mathrm{d}a_k}}{\frac{\mathrm{d}f_{\theta_{\text{old}}}^{-1}(a_k;s_k)}{\mathrm{d}a_k}} \nabla_{\theta'} a \nabla_a Q^{\pi_{\theta_{\text{old}}}}(s, a)|_{a=f_{\theta'}(\epsilon_{\text{reg}};s)} \right.$$

$$\left. \cdot p_{\text{standard}}(\epsilon) \frac{\mathrm{d}f_{\theta_{\text{old}}}^{-1}(a_k; s_k)}{\mathrm{d}a_k} \frac{\mathrm{d}f_{\theta'}(\epsilon_{\text{reg}}; s_k)}{\mathrm{d}\epsilon_{\text{reg}}} \mathrm{d}\epsilon_{\text{reg}} \right]$$

$$= \gamma^k \int_{\epsilon_{\text{reg}}} \left[ \nabla_{\theta'} a \nabla_a Q^{\pi_{\theta_{\text{old}}}}(s, a)|_{a=f_{\theta'}(\epsilon_{\text{reg}};s)} p_{\text{standard}}(\epsilon_{\text{reg}}) \mathrm{d}\epsilon_{\text{reg}} \right],$$

where the product of the derivatives $\frac{\mathrm{d}f_{\theta'}^{-1}(a_k;s_k)}{\mathrm{d}a_k}$ and $\frac{\mathrm{d}f_{\theta'}(\epsilon_{\text{reg}};s_k)}{\mathrm{d}\epsilon_{\text{reg}}}$ is 1, as the functions are inverses. We can clearly see that this expectation matches our target, providing an unbiased estimate.

The extension of this result to the multi-dimensional case is straightforward. Since the source of randomness is a standard Gaussian distribution, each dimension is sampled independently. Consequently, both the overall importance ratio and $p_{\text{reg}}(\epsilon_{\text{reg}})$ factorize into a product of their respective one-dimensional components. Recalling that states are sampled according to $p(s_k = s|\pi_{\theta_{\text{old}}})$, and by moving $\gamma^k$ to the outer integral, we can conclude that this method provides an unbiased estimate of $\int_s \gamma^k p(s_k = s|\pi_{\theta_{\text{old}}}) \int_\epsilon \left[ \nabla_{\theta'} a \nabla_a Q^{\pi_{\theta_{\text{old}}}}(s_k, a)|_{a=f_{\theta'}(\epsilon;s_k)} p_{\text{standard}}(\epsilon) \mathrm{d}\epsilon \right] \mathrm{d}s$.

$\square$

Thanks to proposition 1, by summing over different time step and averaging across different trajectories, we obtain an unbiased off-policy reparameterization gradient estimation.

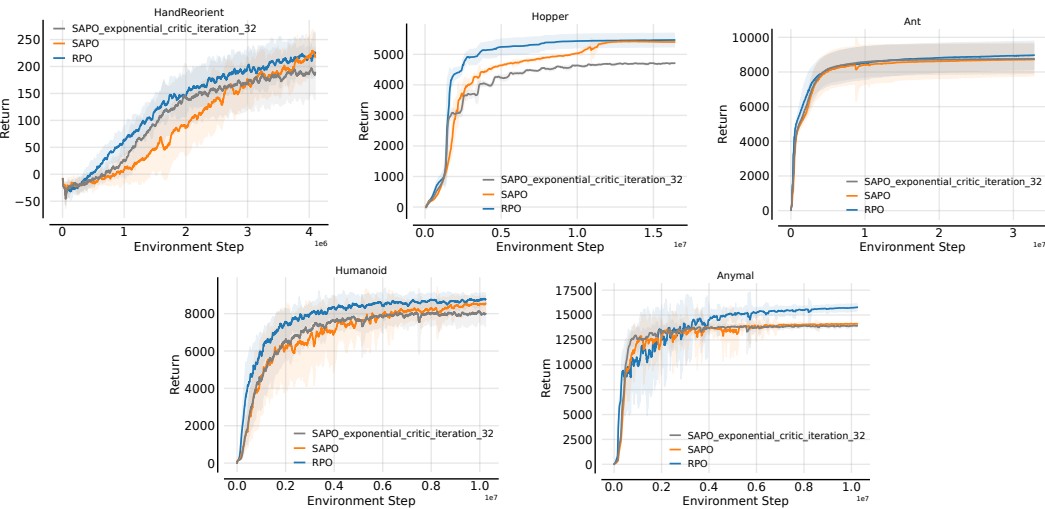

Figure 6: Ablation study for comparison of RPO, SAPO, and SAPO with exponential learning rate schedule and 32 critic update iterations. RPO outperforms SAPO in both settings.

# E  ABLATION EXPERIMENTS ON LEARNING RATE AND CRITIC UPDATE ITERATIONS

RPO uses an exponential learning rate decay and 32 critic update iterations. This is effective because RPO's high sample efficiency allows it to learn quickly in the early stages of training, while the decaying learning rate helps to accelerate final convergence once a good performance level is reached. We investigate the effect of applying the exact same exponential learning rate schedule and 32 critic update iterations to SAPO.

As shown in Figure 6, RPO consistently outperforms SAPO with both settings. Indeed, SAPO's final performance is degraded with an exponential learning rate decay and 32 critic update iterations in most environments. This result indicates that RPO's superior performance stems from its core algorithmic design—namely, the stable sample reuse mechanism—rather than being solely attributable to these specific hyperparameter choices.

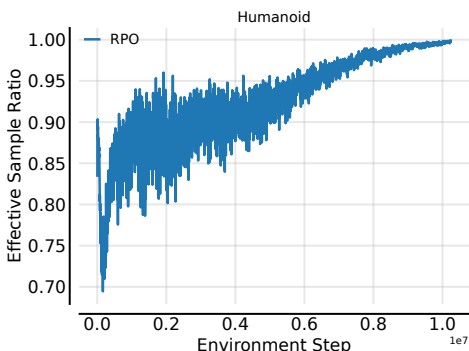

Figure 7: RPO's effective sample ratio on Humanoid Task.

## F  RPO'S EFFECTIVE SAMPLE RATIO

In this section, we analyze the sample utilization efficiency of RPO by measuring the proportion of samples that are not filtered out by the gradient clipping mechanism. We define the *effective sample ratio* as the percentage of samples whose gradients are not zeroed out during the off-policy update epochs (specifically, epochs 2 to 5). Note that in the first epoch, all samples are effective by definition. The results are presented in Figure 7. As shown in the figure, RPO maintains a high effective sample ratio; the lowest observed ratio is approximately $70\%$, which gradually approaches $100\%$ towards the end of training.

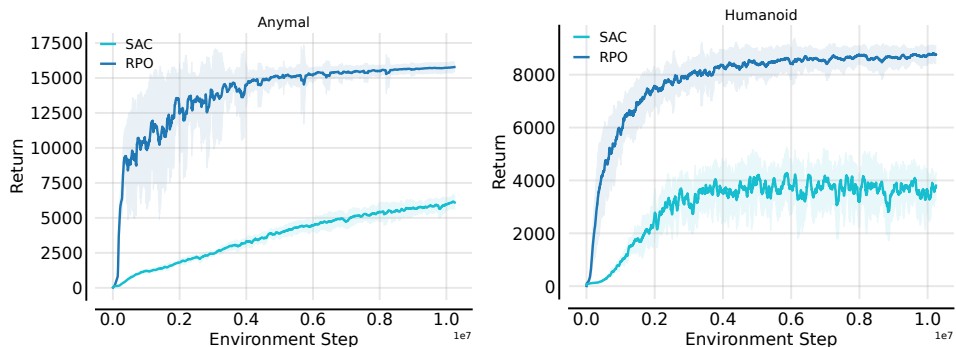

Figure 8: Ablation study for comparison of RPO and SAC.

|                                   | Humanoid | Anymal  |
|-----------------------------------|----------|---------|
| n-step                            | 3        | 10      |
| actor learning rate               | $5e-4$   | $2e-3$  |
| target critic smoothing coefficient | 0.2    | 0.6     |

Table 8: Hyperparameters for SAC.

## G MORE COMPARISONS WITH BASELINES

### G.1 COMPARISON WITH SOFT ACTOR-CRITIC

We compare RPO with Soft Actor-Critic (SAC) (Haarnoja et al., 2018), incorporating the n-step return mechanism for critic training, on two tasks: Anymal and Humanoid. We utilize the implementation from the Mineral repository (https://github.com/etaoxing/mineral) and align the actor and critic MLP hidden layers to $[400, 200, 100]$. We tuned SAC's hyperparameters, including the n-step horizon, target critic smoothing coefficient, and actor learning rate, as detailed in Table 8. The training curves are presented in Figure 8. The results demonstrate that RPO consistently outperforms SAC.

### G.2 COMPARISON WITH PPO TRAINED ON MORE SAMPLES

It is well known that PPO, due to its reliance on REINFORCE-type policy gradients, is much less sample-efficient than RPG-based approaches (Mohamed et al., 2020; Xu et al., 2021). We trained PPO on the Hopper task with 200 million environment steps and on the Anymal task with 100 million environment steps. The results show that even with 10 times more interactions with the environment, RPO still achieves a higher reward than PPO. The training curves are shown in Figure 9, and the performance results are shown in Table 9.

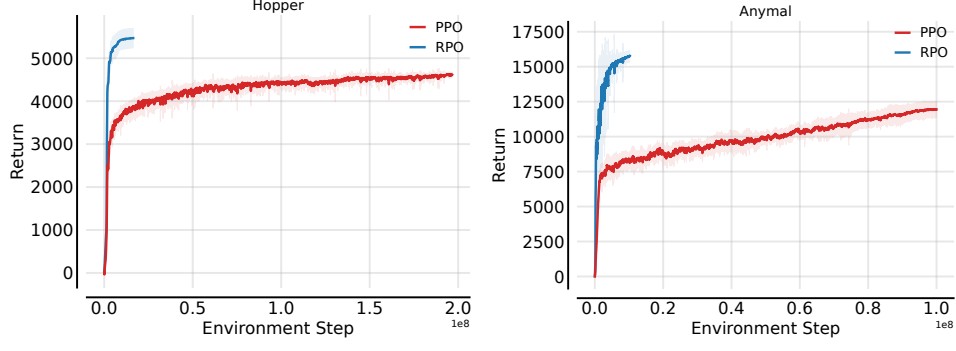

Figure 9: Comparison between RPO and PPO trained on significantly more samples.

|  | Hopper | Anymal |
|---|---|---|
| RPO | $5534.32 \pm 223.68$ | $16006.42 \pm 342.49$ |
| PPO | $4713.93 \pm 44.01$ | $12841.25 \pm 44.01$ |

Table 9: Comparison between RPO and PPO trained on significantly more samples. We report mean and standard deviation for deterministic performance evaluation after training.

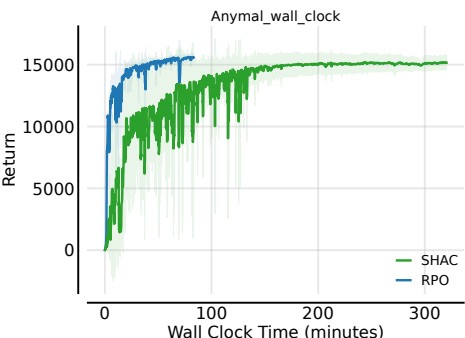

Figure 10: Wall-clock time comparison between RPO (trained for 10 million environment steps) and SHAC (trained for 40 million environment steps).

### G.3 COMPARISON WITH SHAC TRAINED WITH 40 MILLION ENVIRONMENT STEPS

In this section, we compare RPO (trained with 10 million environment steps) against SHAC (trained with 40 million environment steps). We conducted the experiments on a single machine equipped with NVIDIA RTX 4090 GPU. Due to computational constraints, we performed the comparison using three random seeds and recorded the wall-clock time. To ensure a fair comparison with SHAC, we set the number of critic iterations for RPO to 16. Note that RPO achieves comparable performance with either 16 or 32 iterations, as shown in Appendix J. The training curve demonstrates that RPO also holds an advantage in wall-clock time efficiency.

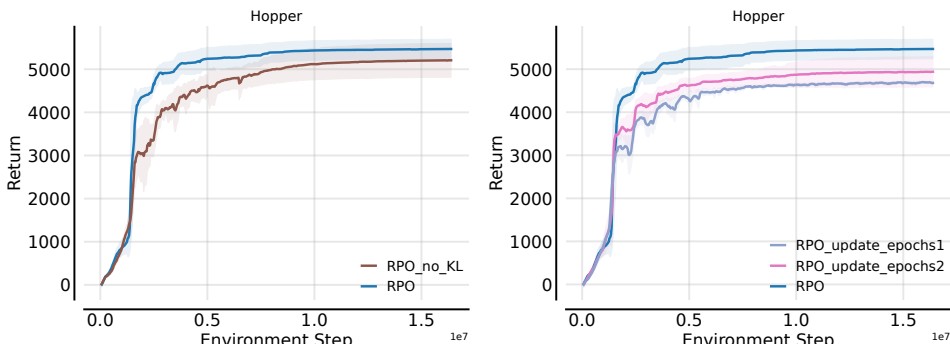

Figure 11: Ablation study for RPO without KL regularization (left) and RPO with 2 policy update epochs (right) on the Hopper task.

## H    MORE ABLATION ON KL DIVERGENCE REGULARIZATION

We conducted further ablation studies on RPO without KL divergence regularization on the Hopper task to isolate its impact. The training curves are presented in Figure 11 (left). As discussed in the main text, RPO without KL regularization learns noticeably more slowly than standard RPO. This retardation is primarily due to unstable policy updates. Notably, the absence of KL regularization also results in a drop in final asymptotic performance.

## I    MORE ABLATION ON SAMPLE REUSE

We extended our ablation studies to evaluate the impact of sample reuse frequency, testing RPO with only 1 or 2 policy update epochs on the Hopper task. The comparative training curves are shown in Figure 11 (right). The results clearly demonstrate that sample reuse is a critical factor for RPO's success. The variant with minimal updates (1 or 2 epoch) significantly underperforms the default multi-epoch setting. This confirms that sample reuse is essential for both maximizing sample efficiency and achieving strong final performance.

| Ablation Component | Parameter Setting | Anymal Return |
|---|---|---|
| *(a) Clipping Values* | $c_{low} = 0.8, c_{high} = 1.0$ | $16006.42 \pm 342.49$ |
| | $c_{low} = 0.8, c_{high} = 0.5$ | $15932.44 \pm 484.60$ |
| | $c_{low} = 0.6, c_{high} = 1.0$ | $16043.71 \pm 376.54$ |
| *(b) Critic Updates* | 32 epochs | $16006.42 \pm 342.49$ |
| | 16 epochs | $15750.03 \pm 617.41$ |
| *(c) Coefficients* | $\lambda_{KL} = 0.2, \lambda_{ent} = 0.25$ | $16006.42 \pm 342.49$ |
| | $\lambda_{KL} = 0.2, \lambda_{ent} = 0.15$ | $15823.76 \pm 317.63$ |
| | $\lambda_{KL} = 0.2, \lambda_{ent} = 0.20$ | $16167.07 \pm 387.03$ |
| | $\lambda_{KL} = 0.15, \lambda_{ent} = 0.25$ | $15983.25 \pm 526.82$ |
| | $\lambda_{KL} = 0.25, \lambda_{ent} = 0.25$ | $16094.55 \pm 305.25$ |

Table 10: Hyperparameter sensitivity analysis on the Anymal task. We report the mean return and standard deviation.

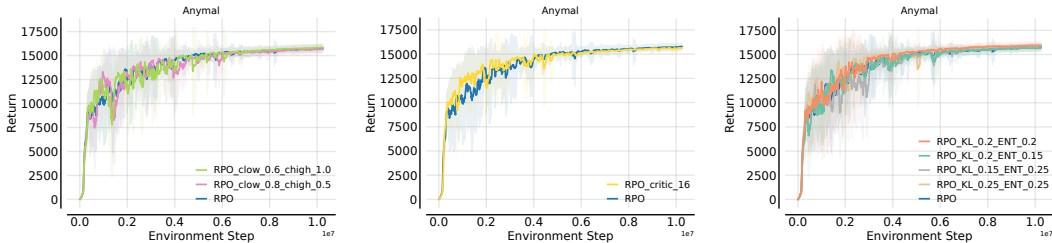

Figure 12: Ablation study on clipping values, critic update epochs, and KL/entropy coefficients.

## J  HYPERPARAMETER SENSITIVITY ANALYSIS

We conducted extensive experiments on the Anymal task to rigorously test RPO's sensitivity to key hyperparameters. Specifically, we evaluated the impact of variations in: (i) the policy gradient clipping bounds ($c_{low}$ and $c_{high}$); (ii) the number of critic update epochs per iteration; and (iii) different combinations of KL regularization ($\lambda_{KL}$) and entropy ($\lambda_{ent}$) coefficients. The corresponding training curves are visualized in Figure 12, and the final performance metrics are detailed in Table 10.

The results demonstrate that RPO exhibits strong robustness across a wide range of hyperparameter settings. Regarding clipping values, both stricter ($c_{high} = 0.5$) and looser bounds yield performance comparable to the default setting, indicating that the algorithm is not brittle to precise clipping thresholds. Similarly, reducing the critic update epochs from 32 to 16 results in similar performance. Finally, RPO can achieve good performance with a range of KL and entropy coefficients.

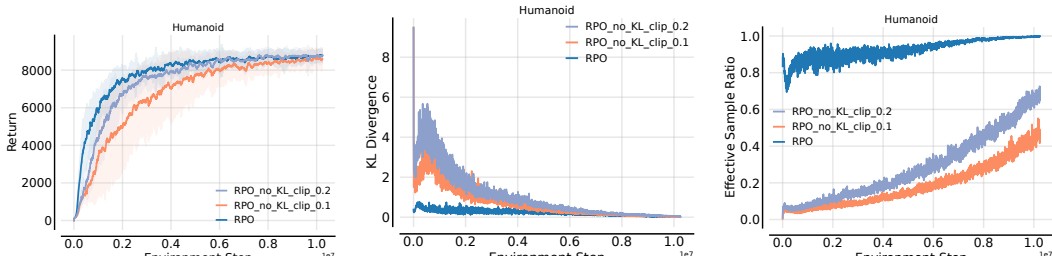

Figure 13: Comparison of return, KL divergence, and effective sample ratios between RPO and RPO without KL regularization (using strict clipping values).

## K DISCUSSION ON THE DESIGN CHOICES OF KL REGULARIZATION AND POLICY GRADIENT CLIPPING

In this section, we discuss the rationale behind using both KL regularization and the policy gradient clipping mechanism to stabilize policy training.

**The clipping mechanism alone is insufficient to stabilize training and reduces the effective sample reuse ratio, whereas KL regularization stabilizes training without sacrificing sample reuse.** To demonstrate this, we conducted experiments on the Humanoid task for RPO without the KL loss, using stricter clipping settings of $c_{low} = 0.1, c_{high} = 0.1$ and $c_{low} = 0.2, c_{high} = 0.2$, respectively. As shown in Figure 13, even with small clipping ranges, policy updates remain unstable, as indicated by the relatively large KL divergence (we report the mean KL divergence metric during training). Furthermore, strict clipping limits the degree of sample reuse. We define the *effective sample ratio* as the percentage of samples whose gradients are not zeroed out during the off-policy update epochs (specifically, epochs 2 to 5). As shown in Figure 13, both small clipping settings result in a very low effective sample ratio during the early phase of training. These limitations translate to a degradation in learning speed, which hurts sample efficiency. On the other hand, KL regularization allows us to explicitly regularize policy updates, leading to stable updates while enabling maximal and stable sample reuse.

**The clipping mechanism is still necessary alongside KL regularization to filter out large importance weight ratios.** As shown in the ablation study in Section 5.3 of the main text, importance weight ratios can be large. Hence, it is natural to incorporate a gradient clipping mechanism to prevent policy updates driven by extreme importance weight ratios and to prevent the probability ratio of certain actions from becoming too low. Additionally, since we must calculate the importance weight ratio for unbiased policy gradient estimation regardless, there is minimal computational overhead in incorporating the policy gradient clipping mechanism.

## L    MORE EXAMPLES FOR INSTABILITY OF RPG-BASED METHODS

In this section, we show examples of unstable seeds for SAPO and SHAC, which are summarized in Figure 14 and Figure 15.

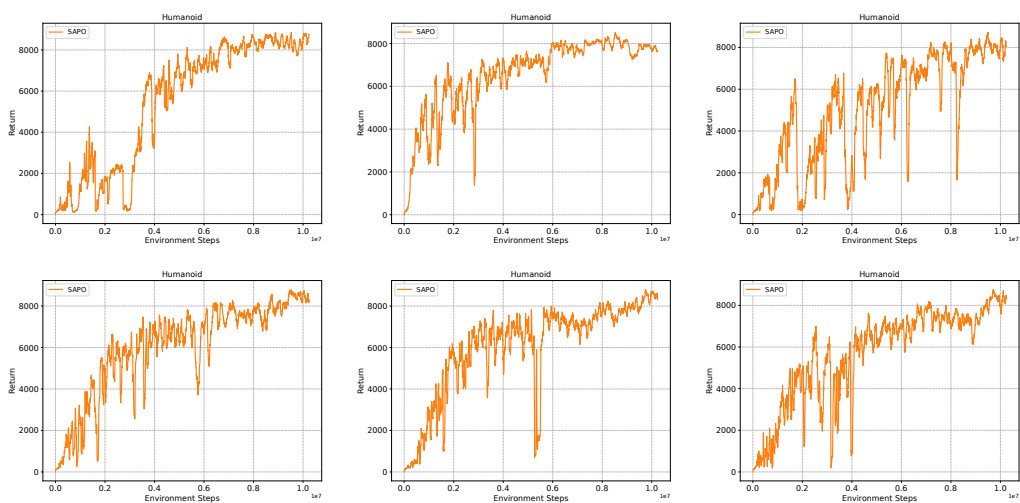

Figure 14: Unstable Seeds for SAPO on the Humanoid tasks.

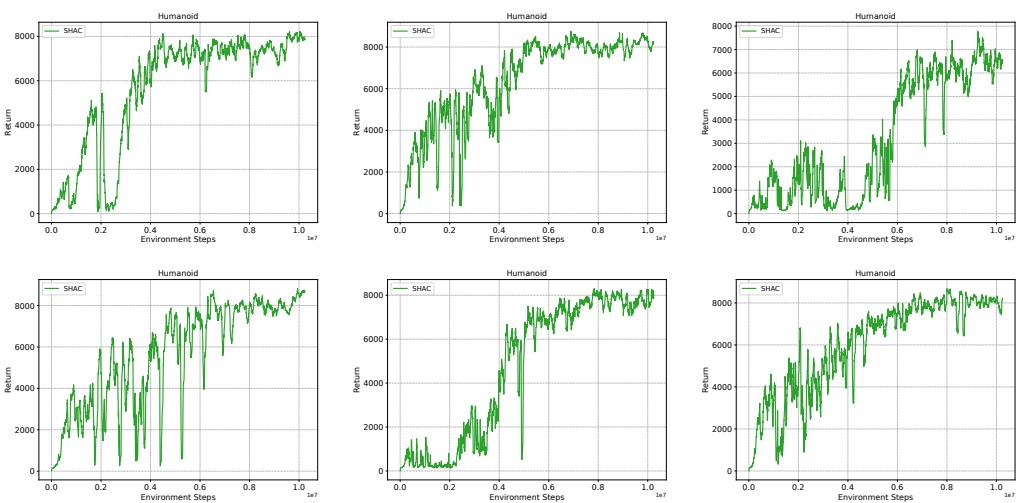

Figure 15: Unstable Seeds for SHAC on the Humanoid tasks.

## M    THE USE OF LARGE LANGUAGE MODELS

In this work, we use large language models for checking and fixing grammar and typos for both main text and appendix.

