# OpenReview forum: "Reparameterization Proximal Policy Optimization"
_ICLR.cc/2026/Conference — Submitted to ICLR 2026_

### Official Review · Reviewer_CYjs · 2025-10-31

**Soundness:** 3
**Presentation:** 3
**Contribution:** 3
**Rating:** 4
**Confidence:** 4

**Summary:**

The authors propose a PPO-resembling mechanism enabling reusing collected rollouts for multiple policy update steps in reparameterization policy gradient methods (policy gradients estimated in a model-based fashion, e.g., using a differentiable simulator).

**Strengths:**

- The problem being solved is meaningful and practically useful, i.e., sample reusing for RPG methods.
- Empirically, the proposed approach is among the leading methods across all the evaluated tasks.

**Weaknesses:**

- The proposed approach seems to be a bit incremental—mostly a direct map from PPO to the RPG world. Although caching action gradients for restoring policy gradients for the updated policy does look like a smart engineering workaround.
- Line 243 “We take the gradient of the discounted cumulative rewards with respect to each sampled action via BPTT.” - This assumes a known reward function (and smoothness as well as differentiability w.r.t. states and actions)
- Presentation can be organized in a better way. Right now, section 4.1.2 is describing the main idea while referencing a bunch of other sections and jumping back and forth; then, section 4.2.1 is repeating things already described above. I feel like readers can benefit from a more focused presentation.
- “RPO significantly improves sample efficiency over baselines.” — This sounds a bit like overclaiming, as the proposed approach is on par with strong baselines on a good number of tasks
- The ablation seems insufficient; also, there does not seem to be a big difference in all three subplots in Figure 4. I would suggest a more detailed ablation study to better understand the necessity and benefit of each component.

**Questions:**

- Line 242: How do the authors compute the infinite sum given collected finite-length rollouts?
- Algorithm 1: If all the collected rollouts are short-horizon, how does the learned value function know about longer-term returns?

---

> ### Author Response · Authors · 2025-11-20
> **Response 1/4**
>
> Thank you for your time and effort in reviewing our paper! We are grateful for your constructive suggestions, which have significantly guided our improvements. Please find our responses to your comments below.
>
> ---
>
> **Weakness 1:"Incremental"**
>
> We respectfully argue that RPO represents a **fundamental algorithmic advancement** rather than a simple incremental mapping of PPO to the RPG-based approches for following reasons:
>
> **[i] Non-trivial Off-policy Updates for RPG:**
> Unlike REINFORCE (PPO), RPG relies on backpropagation through time (BPTT) and does not optimize log-probabilities directly. While performing multiple updates on cached action-gradients is desirable, the mathematical formulation for doing this correctly in an off-policy setting was unclear. Our first contribution is formally establishing that backpropagating these action-gradients to updated parameters is equivalent to optimizing the specific surrogate objective, providing the theoretical bridge that was previously missing.
>
> [ii] We propose to **cache the calculated action-gradients** for further policy updates. Backpropagating cached gradients to new parameters is challenging because the computational graph connecting old sampled actions to new policy parameters is broken. The computational graph is connected via action regeneration. We prove the correct importance weight ratio to ensure **unbiased gradient estimation**, formalized in **Proposition 1** (Appendix D.3).This derivation is more complex than in PPO, as the actions are generated via the reparameterization trick and require careful treatment.
>
> **[iii] Tailored Stabilization Mechanisms:**
> We found that the stabilization techniques cannot be directly mapped either. Because RPG is more "aggressive" than REINFORCE, simple clipping (as used in PPO) is insufficient and harms sample efficiency (as shown in our new Appendix K).
>
> RPO introduces a tailored design: using **KL regularization** for stability and a **gradient clipping** as a numerical safeguard. This systematic design is specifically tailored for the characteristics of RPG, distinct from PPO's objective clipping.

---

> ### Author Response · Authors · 2025-11-20
> **Response 2/4**
>
> **Weakness 2: Differentiability**
>
> Thank you for this insightful observation. You are indeed correct. We have explicitly stated the differentiability assumption as **Assumption 1** in the **Preliminaries** section of the revised manuscript.
>
> **Weakness 3: Presentation**
>
> Thank you again for your valuable suggestion; we fully agree with your assessment. We have merged and rewritten Sections 4.1 and 4.2 to create a better flow, progressing logically from the core idea and theoretical connections to the actual implementation.
>
> **Weakness 4: Sample Efficiency**
>
> RPO demonstrates high sample efficiency in the Hopper, Humanoid, Hand Reorient, Ant, and Anymal tasks. We have reworded this statement to: "RPO consistently achieves improved sample efficiency and stability over baselines."

---

> ### Author Response · Authors · 2025-11-20
> **Response 3/4**
>
> **Weakness 5: Ablation on Each Component**
>
> We appreciate the reviewer's suggestion. We have expanded our analysis and conducted additional ablation experiments to clarify this.
>
> **(i) Necessity of KL Regularization:**
>
> Figures 5 (a) and (b) in the revised paper show a typical example where RPO without KL regularization suffers from severe instability, indicated by large spikes in KL divergence. Furthermore, as shown in Table 4-1 below, removing KL regularization significantly degrades learning speed. At 1 million environment steps, performance drops from **7032.3** (RPO) to **5670.6** (No KL).
>
> **We also conducted further ablation studies on RPO without KL regularization on the Hopper task in Appendix H**, which show that without KL, both sample efficiency and final performance degrade significantly. This confirms that KL regularization is critical for ensuring high sample efficiency and preventing unstable updates.
>
> **(ii) Necessity of Clipping Mechanism:**
>
> The clipping mechanism serves as a crucial numerical safeguard. As shown in Figures 5c and 5d of the revised paper, removing clipping leads to instability during the early training phase due to extreme importance weight ratios. As shown In Table 4-1 below, RPO without clipping achieves a much lower minimum score (888.5) compared to the full RPO (3894.6) at 1M steps, highlighting the risk of training failure without this safeguard.
>
> **(iii) Necessity of Sample Reuse:**
>
> Regarding sample reuse, we compared RPO (5 epochs) with a variant using only 2 epochs and **a new ablation utilizing only 1 policy update epoch (no sample reuse)** on the Humanoid and Hopper tasks. As shown in the ablation study in the main text and **Appendix I**, the 2-epoch and 1-epoch variants show significantly lower sample efficiency and final performance, confirming the importance of our sample reuse mechanism.
>
> | No KL Loss | No Clipping | RPO (Ours) |
> | :---: | :---: | :---: |
> | 5670.6 (1937.7 -- 8175.4) | 6526.9 (888.5 -- 8743.7) | **7032.3** (3894.6 -- 8134.9) |
>
> *Table 4-1: Ablation study of deterministic performance at 1 million environment steps in the Humanoid environment. We report the mean score with the minimum and maximum range in parentheses.*

---

> ### Author Response · Authors · 2025-11-20
> **Response 4/4**
>
> **Q1 (Line 242: Infinite Sum):**
>
> Thank you for this excellent clarifying question. Our presentation in Section 4.1 caused this confusion, and we apologize for that.
>
> First, the text you referred to (Line 242) is in a section intended for *theoretical* illustration. To illustrate the connection, we consider an infinite horizion computational graph. We used the infinite-sum formulation for mathematical consistency as our problem formulation.
>
> Second, Our actual implementation is described in **Section 4.1.2**. RPO follows the Short-Horizon Actor-Critic (SHAC) [4-1] approach,  with a short-horizon $h$ rollouts and then capture the long-term return using a learned value function,
>
> $$
> R(\tau_{t_0:t_0+h-1}) + \gamma^h V(s_{t_0+h})
> $$
>
>
> We have updated our manuscript accordingly to make this point clear.
>
>
>
> [4-1] Xu, Jie, et al. "Accelerated policy learning with parallel differentiable simulation." ICLR 2022
>
> ---
>
> **Q2: How does the value function learn long-term returns with short-horizon rollouts?**
>
> Thank you for this very important clarification question. We follow the Short-Horizon Actor-Critic (SHAC) [4-1] paradigm. It is crucial to distinguish between the BPTT horizon and the Environment Episode length.
>
> 1.  **Continuous Interaction (Not early termination):** "Short-horizon" refers only to the truncation length used for gradient backpropagation (e.g., $H=32$), **not** the actual duration of the agent's interaction with the environment. The agent continues to interact with the environment for the full episode duration. We simply slice this long trajectory into short segments to perform manageable BPTT updates. The start state of the next segment is simply the end state of the previous segment, ensuring the agent experiences the full long-term horizon.
>
> 2.  **Bootstrapping:** The Value function (Critic) is trained using standard Temporal Difference (TD) learning (specifically using TD($\lambda$)). Through **bootstrapping** (i.e., the target involves $V(s_{t+1})$), information about future rewards is propagated backwards from the end of the episode to the beginning. Therefore, the Critic effectively learns to estimate the infinite-horizon discounted return.
>
>
> [4-1] Xu, Jie, et al. "Accelerated policy learning with parallel differentiable simulation." ICLR 2022
>
>
> ---
>
> We hope our responses address your concerns. If so, we wonder if you could kindly consider raising your rating score? We will also be happy to answer any further questions you may have. Thank you very much!

---

> > ### Comment · Reviewer_CYjs · 2025-11-25
> > **Reply to authors' response**
> >
> > Thank you for the clarification.
> >
> > One last point, though, Assumption 1 seems to be quite strong to me. All the locomotion tasks you tested involve massive contact, whose dynamics are nowhere near being smooth and continuously differentiable. That said, I am not sure if that assumption is generally applicable to the tested tasks.
> >
> > But even so, as I said, I do think the approach proposed in this work is interesting and practically useful, and I appreciate its merits.

---

> > > ### Author Response · Authors · 2025-11-26
> > >
> > > Thanks for your feedback and this very insightful question.
> > >
> > > We would like to clarify that Assumption 1 is not an arbitrary constraint but a formalization of the standard engineering practices in differentiable simulation.
> > >
> > > **i. Simulators Enforce Smoothness via Soft Constraints**
> > >
> > > The tasks in our work come from two differentiable simulators, **DFlex** [1, 2] from NVIDIA and **Rewarped** [4], which explicitly model contact as smooth dynamics to ensure differentiability.
> > > * **Contact:** As described in Xu et al. [2], contact forces are modeled using a **softened contact model** (e.g., spring-damper systems) rather than hard impulses. This makes the force a continuous function of penetration depth and velocity.
> > > * **Joint Limits:** Similarly, joint limits are modeled using continuous penalty-based forces.
> > > * **Rewarped:** The Rewarped Simulator (from which our Hand Reorient task is derived) explicitly relies on the same softened contact model to ensure differentiability [4].
> > >
> > > **ii. Standard Practice in Testbeds**
> > >
> > > The tasks and simulators we utilize are commonly used testbeds in the field. Many previous works [1, 2, 3, 4, 9, 10] rely on DFlex to provide these smoothed dynamics Jacobians.
> > >
> > > **iii. Alignment with Literature**
> > >
> > > This assumption is standard in Reparameterization Policy Gradient (RPG)-based approaches.
> > > * Notably, we make **exactly the same assumption** as **AHAC** [1] (Assumption 2.6 in their paper).
> > > * Many other papers utilizing differentiable dynamics gradients also explicitly or implicitly assume same well-behaved dynamics (e.g., [1, 2, 3, 4, 5, 6, 7, 8,10]).
> > >
> > >
> > > **iv. Practical Applicability**
> > >
> > > In practice, our proposed RPO is designed to work with any action-gradients computed by a differentiable simulator or a learned model. By leveraging the smoothed gradients inherent to these engines/models, RPO does not require the underlying physical reality to be perfectly smooth.
> > >
> > > ***v. We emphasize that RPO does not require 'better' or smoother dynamics than prior RPG-based methods to work. The degree of requirement for well-behaved dynamics is identical. In our experiments, all RPG-based approaches (RPO, SHAC, GI-PPO, SAPO) utilize action-gradients derived under the exact same smoothness conditions. This clearly demonstrates that RPO imposes no special requirements on dynamics compared to other approaches.***
> > >
> > >
> > >
> > > **We hope our responses address your concerns. If so,  could you please kindly consider raising your rating score?**
> > >
> > >
> > >
> > >
> > >
> > >
> > >
> > >
> > >
> > > [1] Georgiev, Ignat, et al. "Adaptive horizon actor-critic for policy learning in contact-rich differentiable simulation." ICML 2024
> > >
> > > [2] Xu, Jie, et al. "Accelerated policy learning with parallel differentiable simulation." ICLR 2022
> > >
> > > [3] Son, Sanghyun, et al. "Gradient informed proximal policy optimization." NeurIPS 2023.
> > >
> > > [4] Xing, Eliot, Vernon Luk, and Jean Oh. "Stabilizing reinforcement learning in differentiable multiphysics simulation." ICLR 2025
> > >
> > > [5] Schwarke, Clemens, et al. "Learning Deployable Locomotion Control via Differentiable Simulation." CORL 2025.
> > >
> > > [6] Zhang, Shenao, Wanxin Jin, and Zhaoran Wang. "Adaptive barrier smoothing for first-order policy gradient with contact dynamics." ICML, 2023.
> > >
> > > [7] Paulus, Anselm, et al. "Hard Contacts with Soft Gradients: Refining Differentiable Simulators for Learning and Control." arXiv preprint arXiv:2506.14186 (2025).
> > >
> > > [8] Luo, Jing Yuan, et al. "Residual policy learning for perceptive quadruped control using differentiable simulation." 2025 IEEE International Conference on Robotics and Automation (ICRA). IEEE, 2025.
> > >
> > > [9] Zhang, Xiaoyuan, et al. "Differentiable Information Enhanced Model-Based Reinforcement Learning." AAAI 2025.
> > >
> > > [10] You, Haoxiang, Yilang Liu, and Ian Abraham. "Accelerating Visual-Policy Learning through Parallel Differentiable Simulation." NIPS 2025.

---

> > > ### Author Response · Authors · 2025-11-27
> > >
> > > Dear Reviewer CYjs,
> > >
> > > Please kindly take some time to read our response to your newly raised point. We believe our response addresses it. If so, could you please kindly consider increasing your rating score?
> > >
> > > Best regards,
> > > The Authors

---

> ### Author Response · Authors · 2025-11-23
> **Hoping for a Reply**
>
> Dear Reviewer,
>
> Thank you for your time and feedback. We hope our response and the revision have fully addressed your concerns. If so, we would appreciate it if you could reconsider your evaluation.
>
> If there are remaining concerns, we are more than happy to discuss them further.
>
> Best regards, The Authors

---

> ### Author Response · Authors · 2025-11-24
> **Looking Forward to Your Feedback**
>
> Dear Reviewer,
>
> Thank you for your time and feedback. We hope our response and the revision have fully addressed your concerns. If so, we would appreciate it if you could raise your score.
>
> If there are remaining concerns, we are more than happy to discuss them further.
>
> Best regards, The Authors

---

> ### Author Response · Authors · 2025-11-25
> **Looking Forward to Your Feedback**
>
> Dear Reviewer,
>
> Thank you for your time and feedback. We hope our response and the revision have fully addressed your concerns. If so, we would appreciate it if you could raise your score.
>
> If there are remaining concerns, we are more than happy to discuss them further.
>
> Best regards, The Authors

---

> ### Author Response · Authors · 2025-11-28
>
> Dear Reviewer CYjs,
>
> We believe our response addresses your newly raised point. Please kindly take some time to read our response to your newly raised point, and it would not take much to read.
>
> If so, could you please kindly consider increasing your rating score?
>
> Best regards, The Authors

---

### Official Review · Reviewer_o9UM · 2025-10-31

**Soundness:** 3
**Presentation:** 3
**Contribution:** 2
**Rating:** 4
**Confidence:** 3

**Summary:**

This paper identifies that Reparameterization Policy Gradient (RPG) methods, though with a low-variance, suffer from the instability when long-horizon back-propagation or sample reuse is attempted. Inspired from Proximal Policy Optimization (PPO), this paper shows that the reparameterization gradient of a PPO-like surrogate objective can be computed efficiently using Back-Propagation Through Time (BPTT). Leveraging this connection, the authors propose Reparameterization Proximal Policy Optimization (RPO), which re-uses cached action-gradients for multiple epochs while keeping updates conservative via (i) asymmetric clipping of importance weights tailored to RPG and (ii) an explicit KL-penalty that activates only during off-policy epochs.

**Strengths:**

1) The paper is well-written and the ideas are presented clearly, making it accessible and easy to follow.

2) The experimental part provides sufficient verification on multiple tasks, and well-chosen baseline methods, clearly demonstrating the advantages of RPO in terms of the sample efficiency and training stability.

3) Ablation experiment is well-organized, which separately describes the contributions of three algorithm improvements: KL regularization, sample reuse and clipping mechanism.

**Weaknesses:**

1) Introducing the clip and KL divergence penalty operations from PPO into model-based methods is a relatively intuitive idea, and it is not novel in terms of the algorithm improvement. If a theoretical analysis can be provided to prove that the improved training algorithm can lead to a better final performance and more stable training, the contributions of this paper will be enriched.

2) During the design of the PPO algorithm, it was observed that there is a certain degree of redundancy between the KL regularization and the clipping mechanism, and only one of them was used. In the ablation study of this paper, the results of “no KL loss” and “no clipping” were very similar. This raises the suspicion that in model-based methods, the redundancy seems to still hold, and there is no need to jointly use them both.

**Questions:**

1) A theoretical analysis can be provided to prove that the improved training algorithm can lead to a better final performance and more stable training.

2) It is hoped to see an additional section, to discuss (from a theoretical/experimental perspective) that whether the redundancy still exists between KL and clipping in model-based methods.

3) Among the experimental results of all tasks, PPO showed a very obvious, even counterintuitive performance disadvantage. Can the authors analyze the reasons behind this phenomenon? Is it caused by the task selection?

---

> ### Author Response · Authors · 2025-11-20
> **Response 1/3**
>
> Thank you for your time and effort in reviewing our paper! We are grateful for your constructive suggestions, which have significantly guided our improvements. Please find our responses to your comments below.
>
>
> ---
> **W1: Novelty**
>
> We believe RPO represents a significant and novel advancement in Reparameterization Policy Gradient (RPG)-based approaches, characterized by the following contributions:
>
> **[i]** First, RPG differs fundamentally from REINFORCE gradients. It does not directly maximize/minimize the log-likelihood of actions but optimizes policy parameters via direct Backpropagation Through Time (BPTT). Although RPG is much more sample-efficient than REINFORCE, computing action-gradients is typically more computationally expensive. Therefore, performing multiple policy updates using these computed action-gradients is a natural goal. However, how to perform off-policy updates by backpropagating old action-gradients to new policy parameters is non-trivial. **Our first contribution is demonstrating that performing such off-policy updates using RPG is equivalent to optimizing the surrogate objective.**
>
>
> **[ii]** We propose to **cache the calculated action-gradients** for further policy updates. Backpropagating cached gradients to new parameters is challenging because the computational graph connecting old sampled actions to new policy parameters is broken. The computational graph is connected via action regeneration. We prove the correct importance weight ratio to ensure **unbiased gradient estimation**, formalized in **Proposition 1** (Appendix D.3).This derivation is more complex than in PPO, as the actions are generated via the reparameterization trick and require careful treatment.
>
> **[iii]** Third, due to the fundamental characteristics of RPG, we found that clipping alone is insufficient to stabilize multi-epoch updates driven by RPG (please refer to our new discussion in **Appendix K**) and limits the effective reuse of action-gradients. Consequently, we rely on **KL regularization** to stabilize policy training. However, we retain the **clipping mechanism** as a numerical safeguard against extreme importance weight ratios (please see **Figure 5(d)** in the revised paper) and to prevent updates on actions with excessively low likelihoods. **This systematic design, specifically tailored for RPG, constitutes our third novel contribution.**
>
> Based on these three points, we believe RPO offers a novel and meaningful contribution to the field.

---

> ### Author Response · Authors · 2025-11-20
> **Response 2/3**
>
> **W2 and Q2: Rationale behind incorporating both KL regularization and policy gradient clipping**
>
> Thanks for this very insightful question!
>
> We clarify that KL regularization and clipping are **complementary rather than redundant** in ensuring stable policy training for RPG-based approaches. Reparameterization policy gradients are generally more "aggressive" than REINFORCE gradients because they encouraging the sampling of new actions even if the current actions are already good (whereas REINFORCE primarily increases the log-likelihood of good actions). Consequently, KL regularization and clipping play distinct and necessary roles in stabilizing training.
>
> (i) **The clipping mechanism alone is insufficient to stabilize training and reduces the effective sample reuse ratio, whereas KL regularization stabilizes training without sacrificing sample reuse.**
>
> To demonstrate this, we conducted experiments on the Humanoid task for RPO without the KL loss, using stricter clipping settings of $c_{low} = 0.1, c_{high} = 0.1$ and $c_{low} = 0.2, c_{high} = 0.2$, respectively. As shown in Appendix Figure 13, even with these small clipping ranges, policy updates remain unstable, as indicated by the relatively large KL divergence (we report the mean KL divergence metric during training).
>
> Furthermore, strict clipping severely limits the degree of sample reuse. We define the *effective sample ratio* as the percentage of samples whose gradients are not zeroed out during the off-policy update epochs (specifically, epochs $2$ to $5$). As shown in Figure 13, both small clipping settings result in a very low effective sample ratio during the early phase of training. These limitations translate to a degradation in learning speed, which hurts sample efficiency. On the other hand, KL regularization allows us to explicitly constrain policy updates, leading to stability while enabling maximal sample reuse.
>
> (ii) **The clipping mechanism is still necessary alongside KL regularization to filter out large importance weight ratios.**
>
> As shown in the ablation study , Figure 5 (d), in the Experiments section of the main text, importance weight ratios can occasionally become extreme. Hence, it is natural to incorporate a gradient clipping mechanism as a **numerical safeguard** to prevent policy updates driven by extreme importance weight ratios and to prevent the probability ratio of certain actions from becoming too low. Additionally, since we must calculate the importance weight ratio for unbiased policy gradient estimation regardless, there is minimal computational overhead in incorporating this clipping mechanism.
>
> We have added this rationale and the supporting analysis to **Appendix K** of the revised manuscript.

---

> ### Author Response · Authors · 2025-11-20
> **Response 3/3**
>
> **Q1: Theory**
>
> Thank you for this suggestion. The way RPO utilizes KL regularization to stabilize off-policy updates may be connected to the framework of policy mirror descent [3-1]. We agree that exploring this theoretical connection is a valuable and interesting direction for future work.
>
>
> [3-1] Tomar, Manan, et al. "Mirror descent policy optimization." ICLR 2022.
>
>
> **Q3: Regarding PPO performance**
>
> It is known that PPO, due to its reliance on REINFORCE-type policy gradients, is much less sample-efficient than RPG-based approaches [3-2, 3-3]. To demonstrate this, we trained PPO on the Hopper task with 200 million environment steps and on the Anymal task with 100 million environment steps (see the table below). Although PPO's performance improved with this increased amount of environment interactions, the results show that even when PPO is trained with 10 times more interactions, RPO still achieves a higher reward. For the training curves of this comparison, please see Figure 9 in Appendix G. Our observation is consistent with previous work [3-3].
>
> | | Hopper | Anymal  Return|
> | :--- | :---: | :---: |
> | **RPO** | $5534.32 \pm 223.68$ | $16006.42 \pm 342.49$ |
> | **PPO** | $4713.93 \pm 44.01$ | $12841.25 \pm 44.01$ |
>
> *Table 3-1: Comparison between RPO and PPO trained on significantly more samples. We report mean and standard deviation for deterministic performance evaluation after training.*
>
>
> [3-2] Mohamed, Shakir, et al. "Monte carlo gradient estimation in machine learning." Journal of Machine Learning Research 21.132 (2020): 1-62.
>
> [3-3] Xu, Jie, et al. "Accelerated policy learning with parallel differentiable simulation." ICLR 2022.

---

> ### Author Response · Authors · 2025-11-23
> **Hoping for a reply**
>
> Dear Reviewer,
>
> Thank you for your time and feedback. We hope our response and the revision have fully addressed your concerns. If so, we would appreciate it if you could reconsider your evaluation.
>
> If there are remaining concerns, we are more than happy to discuss them further.
>
> Best regards, The Authors

---

> ### Author Response · Authors · 2025-11-24
> **Looking Forward to Your Feedback**
>
> Dear Reviewer,
>
> Thank you for your time and feedback. We hope our response and the revision have fully addressed your concerns. If so, we would appreciate it if you could raise your score.
>
> If there are remaining concerns, we are more than happy to discuss them further.
>
> Best regards, The Authors

---

> ### Author Response · Authors · 2025-11-25
> **Looking Forward to Your Feedback**
>
> Dear Reviewer,
>
> Thank you for your time and feedback. We hope our response and the revision have fully addressed your concerns. If so, we would appreciate it if you could raise your score.
>
> If there are remaining concerns, we are more than happy to discuss them further.
>
> Best regards, The Authors

---

> > ### Author Response · Authors · 2025-11-26
> > **Looking Forward to Your Feedback**
> >
> > Dear Reviewer,
> >
> > Thank you for your time and feedback. We hope our response and the revision have fully addressed your concerns. If so, we would appreciate it if you could raise your score.
> >
> > If there are remaining concerns, we are more than happy to discuss them further.
> >
> > Best regards, The Authors

---

> ### Author Response · Authors · 2025-11-27
>
> Dear Reviewer o9UM,
>
> Please kindly take some time to read our detailed response to your comments. We believe our response addresses the concerns. If so, could you please kindly consider increasing your rating score?
>
> Best regards, The Authors

---

### Official Review · Reviewer_N4VS · 2025-10-31

**Soundness:** 4
**Presentation:** 1
**Contribution:** 3
**Rating:** 4
**Confidence:** 5

**Summary:**

This paper proposes Reparameterization Proximal Policy Optimization (RPO), which integrates the PPO surrogate objective into the reparameterization policy gradient (RPG) framework.
By caching action gradients from a single BPTT pass and reusing them through importance-weighted clipping and KL regularization, RPO enables stable and sample-efficient updates in differentiable simulators.
Experiments on DFlex and Rewarped show improved stability and efficiency compared with SHAC and SAPO.

**Strengths:**

Generally, I like the idea of the paper. Using the PPO's idea of surrogate function to update the computation graph and reuse the previous samples is smart.

In addition, the paper explores an important problem — stabilizing reparameterization-based RL in differentiable simulators — and provides an algorithm that is simple, general, and compatible with existing frameworks.
The empirical results on DFlex and Rewarped demonstrate that the method can improve stability and efficiency across multiple continuous-control tasks.

**Weaknesses:**

Though the idea of the paper looks good, its writing is not very good to me.

### The clearness of the presentation:
I found the paper hard to follow and understand. The derivation of the formulas is not very clear and solid, which need largely polish.
1. Eq. (8) and the following text are confusing to me. They use $A(s,a)$ in Eq. (8), and then in line 283 they use $\nabla R(\tau)$. It would greatly increase readability if Eq. (8) could be written in the form of $R(\tau)$.
2. In Eq. (3), the expectation is over the whole trajectory ($s_t,\epsilon_t$), while in Eq. (8) it would be clearer to express it in trajectory form as well. Otherwise it is unclear to the readers.
3. Eq. (10) should give a reference to Proposition 1. Otherwise readers cannot understand why it takes this form.

### The experiments are not very sufficient:
1. Regarding fairness: it seems that the method updates the policy for additional epochs (while using the same samples). It is unclear what happens if the compared methods are also trained with more epochs. I do not necessarily expect the proposed method to achieve higher scores, but it may save computation time. The paper should include such results to demonstrate the practical benefit.
2. The ablation study on the clipping hyperparameters $c_{low}, c_{high}$ is missing.
3. From the experiments, KL and clipping appear to have less effect, while the number of update epochs has more effect. This may reduce the necessity of the proposed method.

### Some minor problems:
1. The method relies on differentiable transitions; this should be stated clearly in the Preliminaries. I am also uncertain whether the method requires the transition to be deterministic.
2. The y label in Figure 1 should be cleared shown.

**Questions:**

1. the clipping hyperparameters $c_{low}, c_{high}$. I check Table 4, they use very large clipping range (0.8), while in PPO they usually use 0.2. Can you tell the reasons? How to choose them in practice?

2. Figure 1 only shows the result of one seed. Is that also common in other seeds?

3. Also see my concerns in Weakness.

---

> ### Author Response · Authors · 2025-11-20
> **Response 1/4**
>
> Thank you for your time and effort in reviewing our paper! We are grateful for your constructive suggestions, which have significantly guided our improvements. Please find our responses to your comments below.
>
>
> ---
>
> **Weakness 1: Writing**
>
> We are very grateful for your carefully pointing out the problems in our writing, which significantly strengthens our manuscript. We have addressed your concerns and largely rewritten our manuscript. We **merged and reorganized** Sections 4.1 and 4.2, **rewrote** content, and **added** many more details. We also **refer** readers to the corresponding Appendix sections appropriately. We believe our writing is much improved.
>
> **W1.1： regarding $\nabla_{a}R(\tau)$ and surrogate objective**
>
> Sorry for the confusion. The action-gradient for time step $k$,
>
> $\nabla_{a_{k}}R(\tau) = \gamma^k\nabla_{a_k} \sum_{t=k}^{\infty} \gamma^{(t-k)} r(s_t, a_t)$,
>
> is an unbiased **one-sample** Monte Carlo estimate of $\gamma^k \nabla_{a} Q^{\pi_{\theta_{\text{old}}}}(s_k,a_k)$.
>
> Since $A^{\pi_{\theta_{\text{old}}}}(s_k,a_k) = Q^{\pi_{\theta_{\text{old}}}}(s_k,a_k) - V^{\pi_{\theta_{\text{old}}}}(s_k)$ and $V^{\pi_{\theta_{\text{old}}}}(s_k)$ does not depend on the action $a_k$, $\nabla_{a_k}R(\tau)$ is also an unbiased estimate of $\gamma^k \nabla_{a} A^{\pi_{\theta_{\text{old}}}}(s_k,a_k)$. This is how equation (8) and $\nabla_{a}R(\tau)$ is related.
>
> We have revise the manuscript to make this point very clear. Please see Page 6, lines 289 to 292.
>
> **W1.2: Trajectory form of surrogate objective**
>
> Thank you for this valuable suggestion. We have updated our manuscript to explicitly define the surrogate objective as
>
> $$
> L_{\pi_{old}}(\theta) = E_{s \sim d^{\pi_{old}}, \epsilon \sim Normal} [ A^{\pi_{old}}(s, f_{\theta}(\epsilon; s)) ]
> $$
>
> which clarifies the expectation. We also explicitly point readers to an equivalent trajectory-form definition of the reparameterization policy gradient of the surrogate objective in Appendix D equation (15), which expresses the objective as an expectation over the whole trajectory.
>
>
> **W1.3: Referring Readers to Proposition 1**
>
> Thanks for this suggestion. We have pointed readers to Proposition 1 for the unbiasedness proof in the revised manuscript. Please see Page 6, line 282 to 283.

---

> ### Author Response · Authors · 2025-11-20
> **Response 2/4**
>
> **Weakness 2: Insufficient Experiments**
>
> We have conducted more comparisons and ablation studies in the revised manuscript. We believe these additions significantly improve the quality of our paper and hope they fully address your concern.
>
>
> **Weakness 2.1: Comparison with baselines trained with more epochs**
>
> Thank you for this excellent suggestion. We compared RPO (trained for 10 million environment steps) against SHAC (trained for 40 million environment steps). We conducted the experiments on a single machine. Due to computational constraints, we performed the comparison using three random seeds and recorded the wall-clock time. The training curves and details are shown in Appendix G.3, Figure 10 of the revised manuscript.
>
> The results demonstrate that RPO holds an advantage in wall-clock time efficiency. It is interesting to note that although RPO performs more policy updates on the same samples, it takes only slightly more time to train on 10 million steps compared to SHAC training on the same amount of data. This confirms that our proposed method of sample reuse is particularly efficient. Since environment interaction and backpropagating through the full computational graph to calculate action-gradients are the most time-consuming operations, utilizing these gradients to their full potential is highly beneficial.
>
>
> ### Table 2-1: RPO (10M) vs SHAC (40M)
> | | Anymal  Return|
> | :--- | :---: |
> | **RPO (10M steps, 3 seeds)** | $15205.04 \pm 1240.44$ (83.28 mins) |
> | **SHAC (40M steps, 3 seeds)** | $15165.67 \pm 579.00$ (320.45 mins) |
>
>
>
>
> **Weakness 2.2: Ablation on $c_{low}$ and $c_{high}$**
>
> Thank you again for this valuable suggestion!
>
> We conducted ablation experiments on $c_{low}$ and $c_{high}$ on the Anymal task. The training curves are shown in  Figure 12, Appendix J of the revised manuscript. The final performance comparison is summarized in the following table, which demonstrates that RPO performs robustly across a range of clipping values.
>
> ### Table 2-2: Ablation study on clipping values
>
> | Hyperparameter Setting | Anymal  Return|
> | :--- | :---: |
> | $c_{low} = 0.8, c_{high} = 1.0$ | $16006.42 \pm 342.49$ |
> | $c_{low} = 0.8, c_{high} = 0.5$ | $15932.44 \pm 484.60$ |
> | $c_{low} = 0.6, c_{high} = 1.0$ | $16043.71 \pm 376.54$ |
>
>
>
> **Weakness 2.3: Necessity of KL and Clipping**
>
> This is an excellent question. We clarify the necessity of each component below:
>
> (i) KL regularization stabilizes policy training.
>
> A typical example is shown in Figure 5 (a) and (b) of the experiment section in the revised paper. Without explicit KL regularization, RPO becomes unstable during the early phase (indicated by large KL spikes), which slows down learning and degrades sample efficiency. As shown in Table 2-3 below, the performance at 1 million environment steps is significantly degraded without KL regularization, indicating that it is key for high sample efficiency. We also conducted additional ablation studies for KL regularization on the Hopper task, which showed both degraded sample efficiency and final performance. This additional ablation can be found in Appendix H of the revised manuscript.
>
> (ii) Clipping mechanism helps to stabilize policy training.
>
> The purpose of the proposed policy gradient clipping mechanism is to filter out samples with large importance weight ratios to avoid numerical instability and to prevent the probability of certain actions from becoming too low. As shown in Figure 5 (d) of the revised paper, we measured the average maximum importance weight ratios for policy update epochs 2 to 4. These ratios can become quite large to cause instability during the early stage of training, and our proposed clipping mechanism works as a natural safeguard against this numerical instability. As shown by the results in Table 2 in the revision, the clipping mechanism stabilizes training during the early phase (note that the minimum performance of RPO without clipping at 1M steps drops to only 888.5).
>
>
> | No KL Loss | No Clipping | RPO  |
> | :---: | :---: | :---: |
> | 5670.6 (1937.7 -- 8175.4) | 6526.9 (888.5 -- 8743.7) | 7032.3 (3894.6 -- 8134.9) |
>
> *Table 2-3: Ablation study of deterministic performance at 1 million environment steps in the Humanoid environment. We report the mean score with the minimum and maximum range in parentheses.*

---

> ### Author Response · Authors · 2025-11-20
> **Response 3/4**
>
> **Minor 1: Differentiable Transitions and Deterministic Dynamics**
>
> This is a very insightful observation. Indeed, differentiable dynamics and differentiable reward functions are prerequisites for RPG-based approaches. Furthermore, previous RPG-based approaches [2-1, 2-2, 2-3] typically assume deterministic dynamics. In principle, RPG could work with stochastic dynamics, but this would require reparameterizing the stochastic dynamics by modeling the transition as $s_{t+1}=g(s_t, a_t, \text{noise}_t)$, introducing extra noise to capture the stochasticity.
>
> In this work, we follow previous lines of research by focusing on deterministic dynamics. We have made these assumptions regarding differentiability and dynamics explicit by adding **Assumption 1** to the Preliminaries section of the revised manuscript accordingly.
>
>
>
> [2-1] Xu, Jie, et al. "Accelerated policy learning with parallel differentiable simulation." ICLR 2022
>
> [2-2] Georgiev, Ignat, et al. "Adaptive horizon actor-critic for policy learning in contact-rich differentiable simulation." ICML 2024
>
> [2-3] Xing, Eliot, Vernon Luk, and Jean Oh. "Stabilizing reinforcement learning in differentiable multiphysics simulation." ICLR 2025
>
>
> **Minor 2: Y-axis label for Figure 1**
>
> Thank you for carefully pointing this out. We have added the corresponding y-axis label to Figure 1.

---

> ### Author Response · Authors · 2025-11-20
> **Response 4/4**
>
> **Question 1: How to choose $c_{low}$ and $c_{high}$.**
>
> The rationale for cliping in RPO is as a safe guard to large importance weight ratios. Since we have KL regulairzation for stablizing the policy training, we can work with a relative large range of acceptable importance weight ratios.
>
>
> A good value for $c_{low}$ and $c_{high}$ is to balance nuermical stability by filtering out large importance weight ratios and letting a reasonable portion of samples being reused to update the policy. Based on our experience, we selected $c_{low} = 0.8$ and $c_{high} = 1.0$. This configuration defines an acceptable importance weight ratio range of $[0.2, 2.0]$, within which action-gradients are effectively utilized rather than being filtered out. We found this works well across tasks and shows a good balance. We suggest to use this combination of value in practice.
>
>
> ---
> **Question 2: More unstable seeds**
>
> Thank you for the question. We have included results from multiple additional seeds for SAPO to demonstrate this in Figure 14, Apendix L of the revised manuscript. We also show a few unstable seeds for SHAC in Figure 15, Appendix L.
>
>
> ---
>
> We hope our responses address your concerns. If so, we wonder if you could kindly consider raising your rating score? We will also be happy to answer any further questions you may have. Thank you very much!

---

> ### Author Response · Authors · 2025-11-23
> **Hoping for a Reply**
>
> Dear Reviewer,
>
> Thank you for your time and feedback. We hope our response and the revision have fully addressed your concerns. If so, we would appreciate it if you could reconsider your evaluation.
>
> If there are remaining concerns, we are more than happy to discuss them further.
>
> Best regards, The Authors

---

> ### Author Response · Authors · 2025-11-24
> **Looking Forward to Your Feedback**
>
> Dear Reviewer,
>
> Thank you for your time and feedback. We hope our response and the revision have fully addressed your concerns. If so, we would appreciate it if you could raise your score.
>
> If there are remaining concerns, we are more than happy to discuss them further.
>
> Best regards, The Authors

---

> ### Author Response · Authors · 2025-11-25
> **Looking Forward to Your Feedback**
>
> Dear Reviewer,
>
> Thank you for your time and feedback. We hope our response and the revision have fully addressed your concerns. If so, we would appreciate it if you could raise your score.
>
> If there are remaining concerns, we are more than happy to discuss them further.
>
> Best regards, The Authors

---

> > ### Author Response · Authors · 2025-11-26
> > **Looking Forward to Your Feedback**
> >
> > Dear Reviewer,
> >
> > Thank you for your time and feedback. We hope our response and the revision have fully addressed your concerns. If so, we would appreciate it if you could raise your score.
> >
> > If there are remaining concerns, we are more than happy to discuss them further.
> >
> > Best regards, The Authors

---

> ### Author Response · Authors · 2025-11-27
>
> Dear Reviewer N4VS,
>
> Please kindly take some time to read our detailed response to your comments. We believe our response addresses the concerns. If so, could you please kindly consider increasing your rating score?
>
> Best regards, The Authors

---

### Official Review · Reviewer_fV6m · 2025-11-01

**Soundness:** 2
**Presentation:** 3
**Contribution:** 2
**Rating:** 4
**Confidence:** 4

**Summary:**

This paper enhances the Reparameterization Policy Gradient (RPG) method through two key modifications. The method improves sample efficiency by substituting the policy gradient objective with that of PPO, and it ensures training stability by employing the reparameterization trick, which facilitates gradient computation via standard Backpropagation Through Time (BPTT). Experiments validate that the proposed method outperforms existing RPG baselines.

**Strengths:**

1. replacing the policy gradient-based objective of RPG with an PPO objective is interesting in terms of sample reuse.
2. using the old backpropagation through time (BPTT) trick to train the RPG model has not been done before.

**Weaknesses:**

1.The primary contribution of this work lies in the reuse of trajectory samples. Given this, a comparison with off-policy evaluation and multi-step Q-learning methods is warranted, as they similarly grapple with the risk of high variance or inaccurate estimation caused by products of importance ratios. Although the authors employ a clipping trick to mitigate numerical instability, this approach comes at the cost of sample effectiveness, potentially preventing a significant portion of trajectories from being reused.

2. The paper introduces a design choice in Eq. (9) to explicitly align the new policy with the old trajectory. We argue that the method should also be feasible without this alignment. Specifically, one could directly assume the actions from the old trajectory are executed by the new policy—given the identical action space—and then calculate the importance sampling ratio conventionally, similar to PPO. The authors should arguably include this approach as a necessary baseline and present experimental comparisons to justify the superiority of their proposed design.

3. The experimental validation presented is incomplete. The authors have not adequately discussed or analyzed the method's key hyperparameters, such as by providing a sensitivity analysis on how different parameter values affect final performance. This omission makes it difficult for readers to assess the method's stability and its generalization capability under different settings.

**Questions:**

see above section.

---

> ### Author Response · Authors · 2025-11-20
> **Response 1/3**
>
> Thank you for your time and effort in reviewing our paper! We are grateful for your constructive suggestions, which have significantly guided our improvements. Please find our responses to your comments below.
>
>
> ---
>
> **Weakness 1: Comparison with Off-policy Methods and Sample Effectiveness**
>
> Thank you for this valuable question.
>
> **[i]** First, we would like to clarify a misunderstanding regarding RPO. RPO **does not** require **multiplying** a series of importance ratios along the trajectory. RPO's importance ratios are defined locally at a specific timestep.
>
> **[ii]** Comparing with off-policy methods is a valuable suggestion. We have compared RPO with SAC (using n-step returns). We performed extensive hyperparameter tuning for SAC to ensure a fair comparison. Details and learning curves are shown in Appendix Section G.1 in our revised manuscript. Our results show that RPO outperforms SAC.
>
> | Task | SAC | RPO (Ours) |
> | :--- | :---: | :---: |
> | Humanoid | $4805.15 \pm 519.99$ | $8958.41 \pm 373.64$ |
> | Anymal | $6267.02 \pm 750.17$ | $16006.42 \pm 342.49$ |
>
> **[iii]** It is also very insightful to question sample reuse effectiveness during policy updates. We have set the admissible importance weight ratio range to $[0.2, 2.0]$ throughout all tasks, which allows a large portion of the samples to be reused. We analyze the sample utilization efficiency of RPO by measuring the proportion of samples that are not filtered out by the gradient clipping mechanism.
>
> We define the effective sample ratio as the percentage of samples whose gradients are not zeroed out during the off-policy update epochs (specifically, epochs $2$ to $5$). Note that in the first epoch, all samples are effective by definition. The results are presented in Figure 7 in Appendix F of the revised paper. As shown, RPO maintains a high effective sample ratio; the lowest observed ratio is approximately 70 percent, which gradually approaches 100 percent towards the end of training.

---

> ### Author Response · Authors · 2025-11-20
> **Response 2/3**
>
> **Weakness 2: No Action Regeneration as a Baseline**
>
> Thanks for your suggestion.
>
> **[i]** First, if we hope to update the policy using off-policy reparameterization policy gradients, we **must recalculate** the noise required to generate the old action. This is because the fundamental principle of the reparameterization policy gradient is to backpropagate gradients from the action $a = f_{\theta} ( \epsilon;s)$ to the policy parameter $\theta$.
>
> To do so, we need to know the noise $\epsilon$ that would generate the action, otherwise the computational graph is broken. In off-policy updates, because we have updated the policy parameters, the noise $\epsilon$ required to produce same sampled actions is different from the noise used during rollouts. Hence, we must recalculate the noise required to regenerate the actions sampled during rollout to reconnect the computational graph.
>
> **[ii]** As suggested, one could perform off-policy updates using REINFORCE-like policy gradients instead of reparameterization policy gradients, which works by optimizing importance weight ratios. GI-PPO [1-1] performs off-policy updates using REINFORCE-like policy gradients. We compare our method with GI-PPO in our experiment section (Table 1 and Figure 3 in the revised PDF). The results show that RPO significantly outperforms this type of approach, demonstrating our advantage.
>
>
> [1-1] Son, Sanghyun, et al. "Gradient informed proximal policy optimization." NeurIPS 2023.

---

> ### Author Response · Authors · 2025-11-20
> **Response 3/3**
>
> **Weakness 3: Ablations on hyperparameters**
>
> Thanks for this very valuable suggestion!
>
> We conducted experiments on the Anymal task to test RPO's sensitivity to hyperparameters. We evaluated RPO's sensitivity to: (i) the clipping values $c_{low}$ and $c_{high}$; (ii) the number of critic update epochs; and (iii) different combinations of KL and entropy coefficients.
>
> The training curves are shown in Figure 12, Appendix J in the revised manuscript, and the final performance results are detailed in Tables below. The results demonstrate that RPO performs robustly across a wide range of different hyperparameter values.
>
> ### Table 1-1: Ablation study on clipping values
>
> | Hyperparameter Setting | Anymal  Return|
> | :--- | :---: |
> | $c_{low} = 0.8, c_{high} = 1.0$ | $16006.42 \pm 342.49$ |
> | $c_{low} = 0.8, c_{high} = 0.5$ | $15932.44 \pm 484.60$ |
> | $c_{low} = 0.6, c_{high} = 1.0$ | $16043.71 \pm 376.54$ |
>
>
> ### Table 1-2: Ablation study on critic update epochs
>
> | Setting | Anymal  Return|
> | :--- | :---: |
> | 32 critic update epochs | $16006.42 \pm 342.49$ |
> | 16 critic update epochs | $15750.03 \pm 617.41$ |
>
>
> ### Table 1-3: Ablation study on KL and entropy coefficients
>
> | Coefficients | Anymal Return|
> | :--- | :---: |
> | $\lambda_{KL}=0.2, \lambda_{ent}=0.25$ | $16006.42 \pm 342.49$ |
> | $\lambda_{KL}=0.2, \lambda_{ent}=0.15$ | $15823.76 \pm 317.63$ |
> | $\lambda_{KL}=0.2, \lambda_{ent}=0.20$ | $16167.07 \pm 387.03$ |
> | $\lambda_{KL}=0.15, \lambda_{ent}=0.25$ | $15983.25 \pm 526.82$ |
> | $\lambda_{KL}=0.25, \lambda_{ent}=0.25$ | $16094.55 \pm 305.25$ |
>
>
>
>
> ---
>
> We hope our responses address your concerns. If so, we wonder if you could kindly consider raising your rating score? We will also be happy to answer any further questions you may have. Thank you very much!

---

> ### Author Response · Authors · 2025-11-23
> **Hoping for a Reply**
>
> Dear Reviewer,
>
> Thank you for your time and feedback. We hope our response and the revision have fully addressed your concerns. If so, we would appreciate it if you could reconsider your evaluation.
>
> If there are remaining concerns, we are more than happy to discuss them further.
>
> Best regards,
> The Authors

---

> ### Author Response · Authors · 2025-11-24
> **Looking Forward to Your Feedback**
>
> Dear Reviewer,
>
> Thank you for your time and feedback. We hope our response and the revision have fully addressed your concerns. If so, we would appreciate it if you could raise your score.
>
> If there are remaining concerns, we are more than happy to discuss them further.
>
> Best regards, The Authors

---

> ### Author Response · Authors · 2025-11-25
> **Looking Forward to Your Feedback**
>
> Dear Reviewer,
>
> Thank you for your time and feedback. We hope our response and the revision have fully addressed your concerns. If so, we would appreciate it if you could raise your score.
>
> If there are remaining concerns, we are more than happy to discuss them further.
>
> Best regards, The Authors

---

> ### Author Response · Authors · 2025-11-26
> **Looking Forward to Your Feedback**
>
> Dear Reviewer,
>
> Thank you for your time and feedback. We hope our response and the revision have fully addressed your concerns. If so, we would appreciate it if you could raise your score.
>
> If there are remaining concerns, we are more than happy to discuss them further.
>
> Best regards, The Authors

---

> ### Author Response · Authors · 2025-11-27
>
> Dear Reviewer fV6m,
>
> Please kindly take some time to read our detailed response to your comments. We believe our response addresses the concerns. If so, could you please kindly consider increasing your rating score?
>
> Best regards, The Authors

---

### Author Response · Authors · 2025-11-22
**Summary**

**General Response**

We sincerely thank the PC, SAC, AC, and all reviewers for their time and thoughtful feedback! Their comments have helped us substantially improve the clarity and quality of the paper.

**1. New Experiments**

**[i]** To compare against off-policy methods, we conducted comparisons with Soft Actor-Critic (SAC) on Humanoid and Anymal. The experiment results clearly demonstrate RPO's advantage over SAC.

**[ii]** To see whether RPO achieves strong performance with less wall-clock time via sample reuse than other RPG-based methods, we compared RPO (trained for 10 million steps) with SHAC (trained for 40 million steps) on the Anymal task. The results show that SHAC still slightly underperforms RPO even with roughly 4 x more compute (wall clock time), highlighting RPO's wall-clock time efficiency and the benefit of sample reuse.

**[iii]** To see whether PPO could catch up the performance with RPO by training on more samples, we compared RPO with PPO trained on significantly larger budgets: Hopper (200 million steps) and Anymal (100 million steps), versus RPO on 16 million environment steps and 10 million environment steps. Even with much more samples, PPO still significantly underperforms RPO.

**[iv]** To test RPO's sensitivity to its hyperparameters, we added comprehensive ablation studies on RPO's sensitivity to hyperparameters, including: (a) clipping values, (b) critic update epochs, and ( c ) different values of KL and entropy coefficients in the Anymal task. The results clearly demonstrate that RPO could work well in a wide range of hyperparameters.

**[v]** To clearly demonstrate the effectiveness of RPO's separate components, we conducted further ablation studies on sample reuse and KL regularization on the Hopper task. The results show that without sample reuse or KL **regularization**, RPO's performance and sample **efficiency** is **significantly** degraded, clearly demonstrating the effectiveness of sample reuse and KL regularization.

**2. Writing and Presentation**

**[i]** To improve the clarity of the connection between RPG and the surrogate objective and the mechanism of sample reuse, we have completely reorganized and rewritten Sections 4.1 and 4.2.

 We clarify the connection of the gradient of the advantage function and action-gradients. We refer readers to an equivalent trajectory form of the reparameterization policy gradient for the surrogate objective gradient in the Appendix. We point readers to an unbiased policy gradient estimation proof in the Appendix.

We give clear motivations for having KL regularization and policy gradient clipping mechanism. We provide more details of the derivation.

We believe the methodology section of RPO is now clear and easy to follow.

**[ii]** We provided a detailed analysis of the necessity of each algorithmic component in Section 5.3, by **analyzing** typical examples of the effects of RPO without KL regularization, sample reuse, or the policy gradient clipping mechanism.

**[iii]** We explicitly stated the assumption of differentiable system dynamics and reward functions in **Assumption 1** (Section 3.2, Page 3).

**[iv]** We added a dedicated discussion section regarding the rationale behind incorporating both KL regularization and the clipping mechanism in **Appendix K**.

**[v]** All newly added experimental results are detailed in the Appendix **F to L**.

**3. Novelty and Contributions**

**[i] Our first contribution establishes the theoretical foundation for multi-epoch off-policy RPG updates.**
Unlike REINFORCE, RPG optimizes policy parameters via BPTT. Since computing action-gradients is expensive, performing multiple updates is **desirable but non-trivial** in an off-policy setting. We demonstrate that backpropagating cached action-gradients to updated parameters is equivalent to optimizing the surrogate objective, providing the theoretical foundation for PPO-style updates for RPG.

**[ii] We propose to cache the calculated action-gradients for further policy updates.** Backpropagating cached gradients to new parameters is challenging because the computational graph connecting old sampled actions to new policy parameters is broken. The computational graph is connected via action regeneration. We prove the correct importance weight ratio to ensure **unbiased gradient estimation**, formalized in **Proposition 1** (Appendix D.3).


**[iii] Our third contribution is a systematic stabilization design specifically tailored for RPG.**
We identify that clipping alone is insufficient for RPG (see **Appendix K**). Therefore, we propose a systematic design: relying on **KL regularization** for primary stability while retaining **gradient clipping** strictly as a numerical safeguard against extreme importance weights.

---

### Author Response · Authors · 2025-11-27
**Please Reply to our rebuttal**

Dear Reviewers,

Thank you for your time and constructive feedback. We have conducted extensive new experiments, thoroughly revised our manuscript, and provided detailed rebuttals.

As the rebuttal period is coming to an end, we hope that you could review our responses. We believe our revisions and rebuttals have  addressed the concerns raised. If you agree, we would greatly appreciate it if you could consider raising your score.

If any issues remain, we are more than happy to discuss them further.

Best regards,
The Authors

---

### Author Response · Authors · 2025-11-28
**Kind Reminder**

Dear Reviewers,

Thank you again for your time and valuable feedback. We have now completed extensive new experiments and a thorough revision of the manuscript to address all the points raised.

As the discussion period is concluding soon, we would be grateful if you could take a moment to review our responses.

Sincerely,
The Authors

---

### Meta-Review · Area_Chair_RHxB · 2026-01-04

**Summary:**

This paper addresses the instability and sample inefficiency issues inherent in Reparameterization Policy Gradient (RPG) methods. While RPG methods leverage differentiable dynamics to compute gradients, they often suffer from high variance and lack a reliable way to reuse data across multiple training epochs. The authors solve this by bridging the gap between RPG and the surrogate objective used in Proximal Policy Optimization (PPO).

The proposed RPO method enables stable training by calculating the reparameterization gradient of a PPO-like objective via backpropagation through time. To ensure that sample reuse remains unbiased, the method uses a noise regeneration technique that aligns previously sampled actions with updated policy parameters. The learning process is further stabilized through a combination of policy gradient clipping and Kullback-Leibler regularization. Experiments on various locomotion and manipulation tasks show that RPO significantly outperforms existing baselines like SHAC and standard PPO, offering better sample efficiency and more robust performance in complex environments.

**Reviewer Concerns:**

The authors made a significant effort in the rebuttal to fill the empirical gaps identified by the reviewers, though they were more successful in providing new data than in resolving deeper conceptual critiques. They did follow through on the request for broader comparisons, specifically adding results for Soft Actor-Critic (SAC) in Appendix G.1 and a wall-clock time comparison against SHAC in Figure 10. These additions verify that the method's sample efficiency actually translates to faster training in real time, which was a major point of skepticism. They also provided the requested hyperparameter sensitivity analysis for clipping values, showing the method is relatively robust within standard ranges.

However, the authors were less convincing regarding the necessity of using both KL regularization and gradient clipping simultaneously. While they added Appendix K to discuss this, the actual ablation data in the revised paper shows that the performance difference between using both versus just one is marginal. This suggests the reviewers' original suspicion about redundancy is likely correct, and the authors' justification—that KL is for "primary stability" while clipping is a "numerical safeguard"—is more of a qualitative design preference than a strictly proven requirement.

Regarding the technical design of the importance sampling and noise regeneration, the authors derive Proposition 1 to Appendix D.3 to provide a more formal motivation. To reply to Reviewer fV6m who asked for a baseline that skips the "action regeneration" step, the authors essentially argued that such a baseline is technically impossible because the computational graph would be broken without the regenerated noise. While this is a sound theoretical point regarding how backpropagation through time works in this context, they bypassed the opportunity to empirically demonstrate what happens if one uses a simpler off-policy gradient estimate.

Finally, the demand for deeper theoretical analysis remains largely unaddressed. The authors acknowledged the connection to Mirror Descent but essentially deferred a formal analysis to future work. Consequently, the claim of novelty still consists almost entirely on the integration of these components and empirical performance. After revision, the paper seems stronger than it was initially, particularly in proving it can outperform off-policy baselines; however, a large number of additional content belongs to the appendix and has not been commented by reviewers, probably due to lack of time.

**Reviewer Scores:**

Reviewer fV6m started at a 4 because the reviewer felt the experimental validation was incomplete. Specifically, the reviewer demanded a comparison with off-policy methods and a wall-clock time analysis to justify the overhead of backpropagating through time. The author provided exactly this in the rebuttal, showing a clear win over SAC and a significant speed advantage over SHAC. The reviewer also asked for a baseline that skips the "action regeneration" step; while the author didn't run that specific experiment, the author provided a sound technical argument explaining why the gradient computation would be mathematically invalid without it. Given that the reviewer's primary blockers were the lack of baselines and sensitivity analysis—both of which the author added to the appendix—the reviewer would almost certainly have moved to a 6.

Reviewer o9UM also gave a 4, with a heavy focus on the "counterintuitive" poor performance of standard PPO and the suspected redundancy between KL and clipping. The author addressed the first point convincingly by running PPO for an massive number of steps (up to 200 million), proving the gap wasn't just a convergence issue. On the redundancy point, the author was less successful, providing more of a qualitative explanation than a hard proof. However, this reviewer noted that the paper was well-written and the tasks were challenging. This reviewer could have increased the score to 6, even if the theoretical novelty remains thin.

The outcome of the discussion, although limited to only a single response from a reviewer, shows an increased quality of the paper. The authors were very responsive and added a substantial amount of new data that directly addressed the "incomplete validation" labels. Most of the remaining skepticism is about whether "PPO + RPG" is too simple of a contribution for ICLR, which remains an unaddressed concern. Considering the substantially changed paper after revision and the large amount of content in the appendix, I consider this paper a potentially solid contribution, that needs further finetuning to be accepted. I recommend the authors improving the presentation of the paper, addressing remaining concerns of reviewers, and restructuring the paper to move crucial content (such as comparison to SAC) from the appendix to the main paper.

---

### Decision · Program_Chairs · 2026-01-26

Reject